# FANCD2 modulates the mitochondrial stress response to prevent common fragile site instability

Philippe Fernandes[1], Benoit Miotto [2], Claude Saint-Ruf[2], Maha Said[1], Viviana Barra [1,3], Viola Nähse[4], Silvia Ravera[5], Enrico Cappelli[6] & Valeria Naim [1✉]

Common fragile sites (CFSs) are genomic regions frequently involved in cancer-associated rearrangements. Most CFSs lie within large genes, and their instability involves transcription- and replication-dependent mechanisms. Here, we uncover a role for the mitochondrial stress response pathway in the regulation of CFS stability in human cells. We show that FANCD2, a master regulator of CFS stability, dampens the activation of the mitochondrial stress response and prevents mitochondrial dysfunction. Genetic or pharmacological activation of mitochondrial stress signaling induces CFS gene expression and concomitant relocalization to CFSs of FANCD2. FANCD2 attenuates CFS gene transcription and promotes CFS gene stability. Mechanistically, we demonstrate that the mitochondrial stress-dependent induction of CFS genes is mediated by ubiquitin-like protein 5 (UBL5), and that a UBL5-FANCD2 dependent axis regulates the mitochondrial UPR in human cells. We propose that FANCD2 coordinates nuclear and mitochondrial activities to prevent genome instability.

---

[1] CNRS UMR9019, Université Paris-Saclay, Gustave Roussy, Villejuif, France. [2] Université de Paris, Institut Cochin, INSERM, U1016, CNRS, UMR8104, Paris, France. [3] Department of Biological, Chemical and Pharmaceutical Sciences and Technologies, University of Palermo, Palermo, Italy. [4] Department of Molecular Cell Biology, Institute for Cancer Research, The Norwegian Radium Hospital, Oslo, Norway. [5] Experimental Medicine Department, University of Genoa, Genoa, Italy. [6] Haematology Unit, Istituto Giannina Gaslini, Genoa, Italy. ✉email: valeria.naim@gustaveroussy.fr

Common fragile sites (CFSs) are genomic regions that are prone to form breaks and gaps within metaphase chromosomes in response to replication stress[1]. CFSs are known to drive genomic instability from the earliest steps of tumor development[2,3]. CFS instability is cell-type dependent and is influenced by the cell replication and transcription programs[4–6]. The transcription of very large genes encompassing CFSs can conflict with replication[4], modify their replication dynamics[7] and promote the formation of R-loops and secondary structures that cause fork stalling[8], leading to their incomplete replication when cells enter mitosis. Incomplete replication of CFSs leads to the persistence of late replication intermediates that are processed by structure-specific endonucleases, inducing mitotic defects and genomic instability if not properly resolved in a timely manner[9–12]. Despite their intrinsic instability, CFSs and their associated genes are evolutionarily conserved, suggesting that CFSs may function as sensors of cellular stress[13,14].

Among DNA replication and repair proteins, members of the FANC pathway (encoded by the *FANC* genes) function as master regulators of CFS maintenance[15]. The FANC pathway is dysfunctional in individuals with Fanconi anemia (FA), a rare chromosome instability disorder characterized by bone marrow failure, predisposition to acute myeloid leukemia and epithelial cancers, and hypersensitivity to DNA interstrand crosslinks (ICLs) and endogenous aldehydes[16–18]. Chromosomal aberrations in FA patients occur preferentially at CFSs[19–21]. FANCD2, a key component of the FANC pathway, has been shown to relocalize to large genes encompassing CFSs after replication stress[22,23] and form foci at CFSs during mitosis, where it cooperates with the Bloom's syndrome helicase (BLM) to prevent chromosomal abnormalities[24,25]. In vivo, CFS instability can occur following physiological replication stress and is associated with impaired karyokinesis and megakaryocyte differentiation in *Fanca−/−* mice[26]. The FANC pathway is involved in coordinating replication and transcription by preventing or resolving R-loops[27,28] and FANCD2 has been shown to promote CFS replication by limiting R-loop formation[29]. Therefore, failure to prevent or resolve R-loops and transcription-associated replication stress and DNA damage may be the cause of the genomic instability that underlies the cancer predisposition of FA patients.

In addition to their nuclear functions, FANC proteins have been shown to play non-canonical roles in the regulation of mitochondrial function and redox metabolism[30]. FANCD2 regulates mitochondrial energy metabolism by interacting with ATP5a[31], and Fancd2 has been shown to interact with components of the mitochondrial nucleoid and to regulate mitochondrial gene transcription and translation in vivo[32,33]. In addition, FANC proteins regulate mitophagy by interacting with PARK2[34], the product of the *PRKN* gene (also known as *Parkin*) encompassing the CFS FRA6E, which is mutated in Parkinson disease and involved in mitochondrial quality control[35]. Mitochondrial dysfunction is an important effector of the FA cellular and clinical phenotype[36,37], as tumor incidence and the hematopoietic defects in *Fanc*-deficient mice can be improved by antioxidant treatments[38,39]. However, whether these two independent functions in mitochondrial homeostasis and genome stability are mechanistically linked is unclear.

Mitochondria regulate many aspects of cellular metabolism, including energy production and nucleotide and amino acid metabolism. They are bounded by a double membrane system with four distinct functional compartments—the outer membrane, the inner membrane, the intermembrane space, and the matrix. Maintenance of the protein-folding environment within each compartment is required for proper organelle function[40]. The components of the respiratory chain complexes required for oxidative phosphorylation (OXPHOS) activity are encoded by both mitochondrial and nuclear genomes, and coordinated expression from both genomes is crucial to allow the stoichiometric assembly and function of these complexes[41]. Defective import, folding, or assembly of these complexes is sensed by mitochondrial protein quality control systems that activate a feedback signaling pathway, dubbed the mitochondrial unfolded protein response (mtUPR), in order to recover mitochondrial homeostasis[42]. Similarly, the cytosol and endoplasmic reticulum (ER) are exposed to nascent polypeptides and require dedicated protein-folding machinery. To adjust folding capacity and proteostasis, eukaryotic cells have evolved organelle-specific UPRs[43]. The common principles of the UPRs are the dynamic activation of signal transduction pathways involving transient attenuation of protein synthesis and load, and a transcriptional response that increases organelle capacity for handling unfolded proteins, allowing metabolic adaptation. In cases of prolonged or excessive UPR activation, in which homeostasis cannot be re-established, UPR signaling drives cell death.

In the present study, we demonstrate a role for FANCD2 in the mitochondrial stress response that links mitochondrial dysfunction with genome instability. We show that CFS gene transcription is dependent on mitochondrial activity and is induced by mitochondrial stress. FANCD2 depletion induces mitochondrial dysfunction, activation of the mitochondrial stress response and CFS gene expression, leading to CFS instability, while attenuation of OXPHOS metabolism decreases CFS gene transcription and rescues chromosome fragility. Mitochondrial stress induces CFS gene transcription and promotes FANCD2 relocalization to CFS genes. FANCD2 binding to CFS is dependent on CFS gene transcription and increases in a dose-dependent manner. In addition, we show that FANCD2 is dispensable for maintaining CFS stability in the absence of transcription. The induction of CFS genes is mediated by ubiquitin-like protein 5 (UBL5), which is involved in mtUPR signaling in *C. elegans*, and interfering with this pathway partially restores chromosome stability. We propose that CFSs are part of a metabolic checkpoint, and by tuning the mitochondrial stress response with CFS replication, FANCD2 promotes metabolic homeostasis and genome integrity.

## Results

**FANCD2 attenuates CFS gene expression to promote CFS stability.** To analyze the role of FANCD2 in CFS gene expression and stability we used a model cell line, HCT116, in which CFSs have been characterized[44]. Knockdown of FANCD2 increased the transcription of all mapped large CFS genes, whereas the expression of *PTPRG*, a large non-CFS gene close to the CFS-encompassing *FHIT* gene in the FRA3B region, was unchanged (Fig. 1a). The increased CFS gene transcription in FANCD2-depleted cells was associated with increased CFS instability, as measured by fluorescence in situ hybridization (FISH) using probes to detect breaks at the *FHIT*/FRA3B and *PARK2*/FRA6E loci (Fig. 1b, c). We verified upregulation of FHIT expression at the protein and RNA levels, the latter by measuring nascent *FHIT* RNA transcripts using 5-ethynyl uridine (EU) (Fig. 1d). Increased FHIT expression was confirmed using two independent siRNAs targeting FANCD2 (Fig. 1e); upregulation was also observed after FANCD2 depletion in HeLa and RKO cells and, to a lesser extent, after downregulation of the FANC core protein FANCA and of the FANCD2-interacting partner FANCI (Fig. 1f–i).

To determine the role of transcription in CFS instability and in FANCD2 function, we deleted the *FHIT* promoter in HCT116 cells using CRISPR/Cas9 editing and verified that *FHIT* transcription was suppressed (*FHIT*-KO, Fig. 2a). We assayed FRA3B instability by analyzing metaphase spreads for the frequency of FRA3B breakage, using FISH, after treatment with

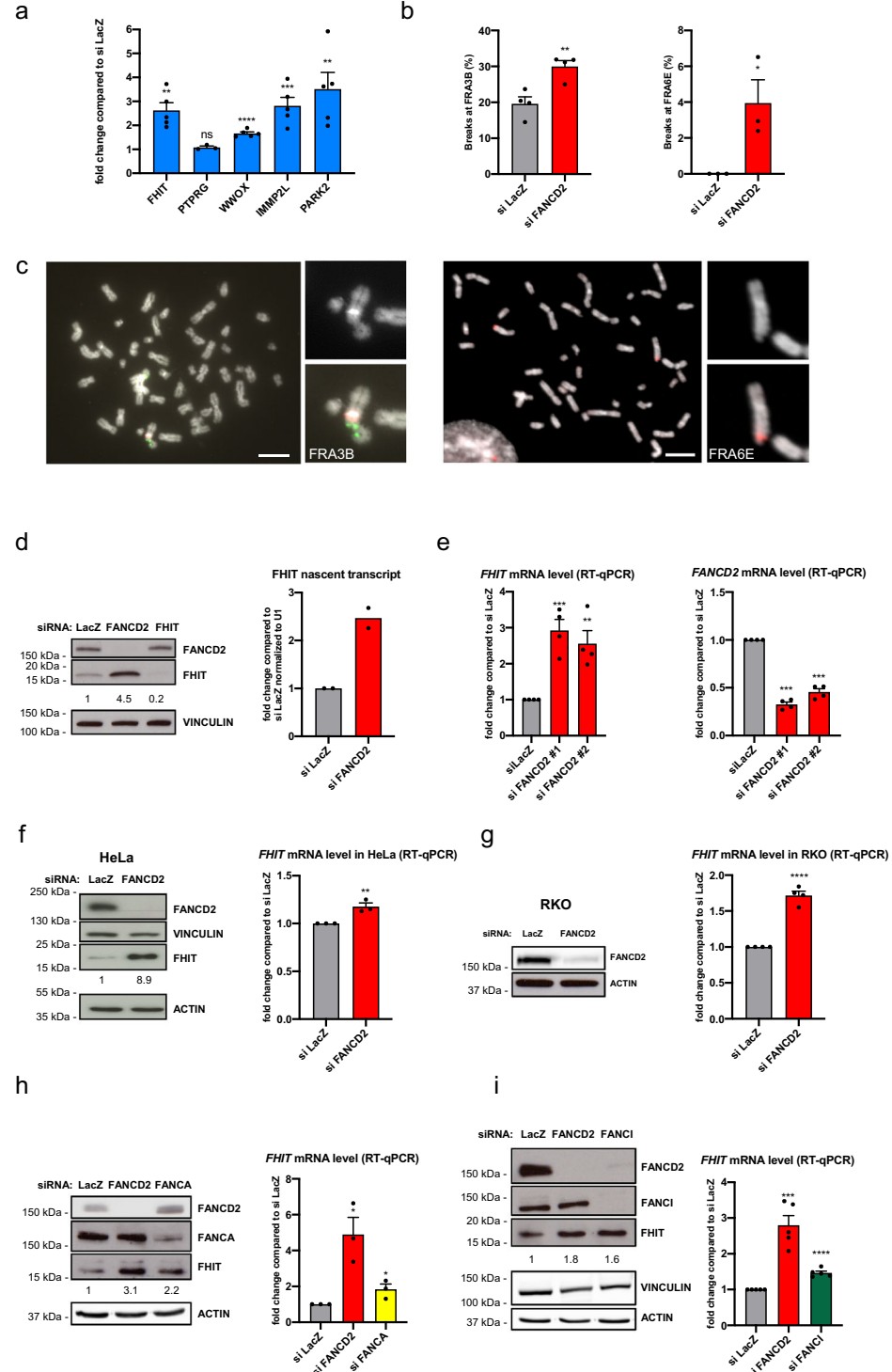

low doses of aphidicolin (APH), which specifically induces CFS instability[45]. FRA3B breaks were significantly reduced in *FHIT*-KO cells compared to parental (*FHIT* wt) cells (Fig. 2b), consistent with transcription inducing CFS instability. Residual breaks at FRA3B that formed in control (siLacZ) cells in the absence of *FHIT* transcription (7.82%) were not increased by FANCD2 depletion (4.06%), demonstrating that the role of FANCD2 at CFSs is linked to transcription of the corresponding gene. Consequently, we examined whether FANCD2 binding to CFSs was dependent on CFS gene transcription. FANCD2 ChIP followed by qPCR in wt and *FHIT*-KO cells revealed that

FANCD2 binding to *FHIT* was substantially reduced in the absence of transcription, whereas binding to other CFS genes was not affected (Fig. 2c). Therefore, FANCD2 is targeted to CFS genes and prevents their fragility in part in a transcription-dependent manner.

**FANCD2 targets mitochondrial UPR-response elements**. To identify regulatory sequences modulating CFS gene transcription and FANCD2 function, we analyzed FANCD2 genomic binding sites by chromatin immunoprecipitation sequencing (ChIP-seq)

**Fig. 1 FANCD2 depletion induces CFS gene expression. a** mRNA levels of large CFS genes measured by RT-qPCR after siFANCD2 transfection, compared to levels after siLacZ transfection. $n = 5$ (FHIT, WWOX, IMMP2L, PARK2), $n = 3$ (PTPRG) independent experiments. **$p = 0.0011$ (FHIT), ****$p < 0.0001$, ***$p = 0.0009$, **$p = 0.0067$ (PARK2). **b** Frequency of FRA3B and FRA6E breakage presented as the percentage of chromosome 3 and chromosome 6 homologs, with breaks at FRA3B and FRA6E, respectively. $n = 4$ and $n = 3$ independent experiments for FRA3B and FRA6E breakage, respectively. **$p = 0.0064$, *$p = 0.0377$. **c** Left, example of FISH analysis of metaphase spread from siFANCD2-transfected cells after treatment with 0.3 μM APH stained for DNA (DAPI, grayscale), a centromeric probe for chromosome 3 (red) and a *FHIT*/FRA3B FISH probe (green). Right, example of FISH analysis of metaphase spread from siFANCD2-transfected cells after treatment with 0.3 μM APH stained for DNA (DAPI, grayscale), and a *PARK2*/FRA6E FISH probe (red). Scale bars, 10 μM. **d** Western blot of whole-cell lysate of control, FANCD2, and FHIT siRNA-transfected HCT116 cells showing the increased FHIT protein level in FANCD2-depleted cells (left). Cells were transfected with siRNA against *FHIT* as a specificity control for the FHIT antibody. The relative increase of FHIT protein level is reported under the corresponding band. Quantification of nascent EU-labeled *FHIT* transcripts after control or FANCD2 siRNA transfection by RT-qPCR normalized to *U1* (*RNU1-1*) RNA gene expression (right). $n = 2$ independent experiments. **e** RT-qPCR analysis of FHIT expression using two independent FANCD2 siRNAs (left). FANCD2 downregulation was estimated by RT-qPCR (right). $n = 4$ independent experiments. ***$p = 0.0008$, **$p = 0.0048$. **f** Western blot of whole-cell lysate of control and FANCD2 siRNA-transfected HeLa cells (left). The relative increase of FHIT protein level is reported under the corresponding band. mRNA levels of *FHIT* measured by RT-qPCR after treatment with control or FANCD2 siRNA (right). $n = 3$ independent experiments. **$p = 0.0089$. **g** Western blot of whole-cell lysate of control and FANCD2 siRNA-transfected RKO cells (left). Note that RKO cells expressed very low levels of FHIT and that FHIT protein was not detected by Western blot in this cell line. Quantification of *FHIT* mRNA levels by RT-qPCR after treatment of cells with control or FANCD2 siRNA (right). $n = 4$ independent experiments. ****$p < 0.0001$. **h** FHIT expression increases after FANCD2 and FANCA depletion in HCT116 cells relative to the control, observed at the protein level by Western blot (left) and the mRNA level by RT-qPCR (right). The relative increase of FHIT protein level is reported under the corresponding band. $n = 3$ independent experiments. *$p = 0.0147$ (siFANCD2), *$p = 0.0459$ (siFANCA). **i** FHIT expression increases after FANCD2 and FANCI depletion in HCT116 cells relative to the control, observed at the protein level by Western blot (left) and the mRNA level by RT-qPCR (right). The relative increase of FHIT protein level is reported under the corresponding band. $n = 5$ independent experiments. ***$p = 0.0002$, ****$p < 0.0001$. Error bars are standard error of the mean (SEM).

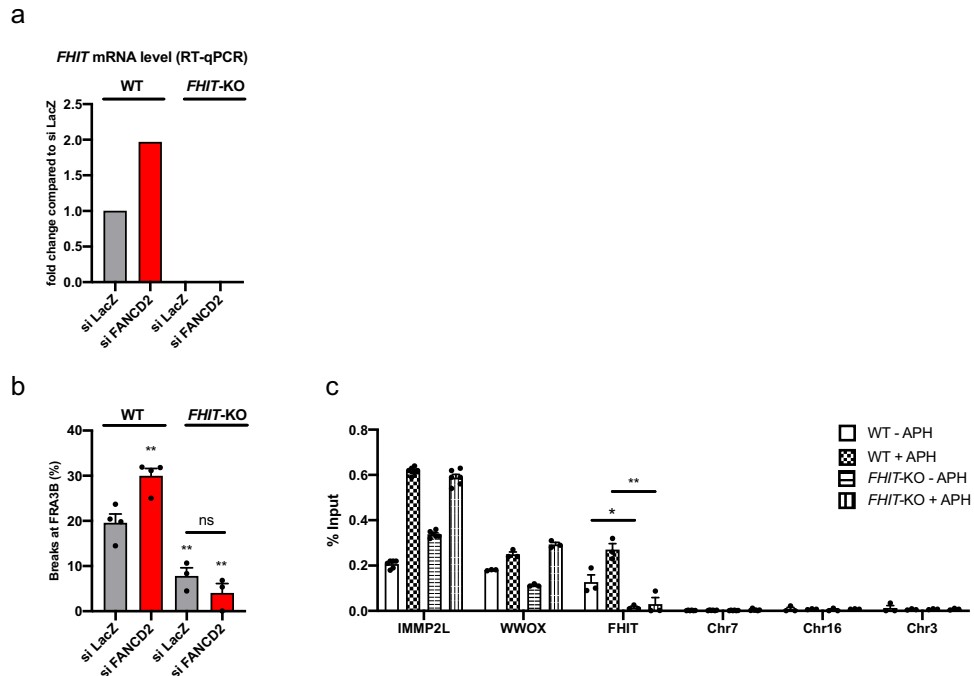

**Fig. 2 FANCD2 binds to CFS genes and prevents their instability in a transcription-dependent manner. a** FHIT mRNA level detected by RT-qPCR in WT and *FHIT*-KO cells after control or FANCD2 siRNA transfection. **b** FRA3B instability in WT and *FHIT*-KO cells transfected with control or FANCD2 siRNA and treated with 0.3 μM APH. $n = 4$ (WT), $n = 3$ (*FHIT*-KO) independent experiments. **$p = 0.0064$ (WT_siFANCD2), **$p = 0.0074$ (*FHIT*-KO_siLacZ), **$p = 0.0028$ (*FHIT*-KO_siFANCD2). **c** FANCD2 ChIP followed by qPCR in wild-type (WT) and *FHIT*-KO cells treated or not with 0.3 μM APH. The results are expressed as the percentage of input. $n = 5$ (IMMP2L), $n = 3$ (WWOX, FHIT). *$p = 0.026$, **$p = 0.0035$. Error bars are standard error of the mean (SEM).

of the endogenous protein in samples that were either untreated or had undergone replicative stress induced by low doses of APH, which induces FANCD2 accumulation and persistence at CFS[22,24]. In both conditions, FANCD2 peaks were enriched in genic regions compared with randomly located peaks (Supplementary Fig. 1a). We analyzed the regions with high levels of FANCD2 after APH treatment and observed that they were enriched in large loci (Supplementary Fig. 1b), the majority of which corresponded to previously characterized CFSs (Table 1),

in agreement with the results in human U2OS and chicken DT40 cells[22,23]. FANCD2 was also recruited to some CFS genes, such as *FHIT* and *WWOX*, under the untreated condition (Fig. 3a). Most of these genes have function in mitochondrial activity, ER dynamics, and secretory pathways (Table 1). We also found that FANCD2 enrichment at CFS genes increased with expression levels (Supplementary Fig. 1c).

A bioinformatic analysis to identify DNA motifs enriched at FANCD2 genomic binding sites revealed a significant enrichment

**Table 1 Genes most enriched in FANCD2 after APH treatment.**

| Rank | Gene ID | Genomic location/fragile site | Gene size (bases) | Description/function |
|---|---|---|---|---|
| 1 | IMMP2L | 7q31.1 FRA7K | 899,887 | Inner mitochondrial membrane peptidase |
| 2 | DOCK1 | 10q26.2 FRA10F | 547,109 | Dedicator of cytokinesis 1, guanine nucleotide exchange factor, cell motility |
| 3 | EXOC4 | 7q33 FRA7H | 813,523 | Exocyst complex component 4, exocyst component, vesicle transport |
| 4 | FHIT | 3p14.2 FRA3B | 1,503,873 | Fragile histidine triad, nucleotide metabolism, mitochondrial $Ca^{2+}$ uptake |
| 5 | SPATA17 | 1q41 | 240,373 | Spermatogenesis associated 17, calmodulin binding |
| 6 | PTPRG | 3p14.2 (FRA3B) | 736,045 | Protein tyrosine phosphatase, receptor type G |
| 7 | SMYD3 | 1q44 FRA1I | 758,003 | KMT3E, histone-lysine N-methyltransferase |
| 8 | DPYD | 1p21.3 FRA1E | 843,317 | Dihydropyrimidine dehydrogenase |
| 9 | PLCB1 | 18q12.2 FRA18A[a] | 891,110 | Phospholipase C beta 1, intracellular transduction |
| 10 | DCDC1 | 11p13 FRA11E | 539,442 | Doublecortin, Golgi-derived vesicle transport |
| 11 | GRID1 | 10q23.2 putative CFS[6,23] | 766,939 | Glutamate receptor, ionotropic, delta 1, synaptic plasticity |
| 12 | FARS2 | 6p25.1 FRA6B | 510,566 | Phenylalanyl-TRNA synthetase 2, mitochondrial, mitochondrial translation |
| 13 | RABGAP1L | 1q25.1 FRA1G | 835,899 | GTP-hydrolysis activating protein (GAP) for small GTPase RAB22A |
| 14 | PGCP | 8q22.1 FRA8B | 504,428 | Carboxypeptidase Q, aminopeptidase |
| 15 | SHANK2 | 11q13.4 FRA11H?[a] | 784,883 | SH3 and multiple ankyrin repeat domains 2, synaptic transmission |
| 16 | PARD3B | 2q33.3 FRA2I | 1,074,370 | Par-3 family cell polarity regulator beta, asymmetrical cell division and cell polarity |
| 17 | NBEA | 13q13.3 | 730,736 | Neurobeachin, lysosomal-trafficking Regulator 2 |
| 18 | FHOD3 | 18q12.2 FRA18A[a] | 482,364 | Formin homology 2 domain containing 3, interaction with SQSTM1 |
| 19 | SLCO3A1 | 15q26.1 | 318,741 | Solute carrier organic anion transporter family member 3A1, |
| 20 | XYLT1 | 16p12.3 putative folate sensitive FS[b] | 369,113 | Xylosyltransferase 1, glycosaminoglycan metabolism in the ER |
| 21 | GPATCH2 | 1q41 | 204,111 | G-patch domain containing 2, spermatogenesis, |
| 22 | MAD1L1 | 7p22.3 | 417,452 | Mitotic arrest deficient 1 like 1 |
| 23 | PRKG1 | 10q11.23 FRA10G?[a] | 1,307,200 | Protein kinase, CGMP-dependent, Type I, mediator of the nitric oxide (NO)/cGMP signaling pathway |
| 24 | MGMT | 10q26.3 | 300,859 | O-6-methylguanine-DNA methyltransferase, DNA repair |
| 25 | FOXP1 | 3p13 | 629,297 | Forkhead Box P1, transcription factor |
| 26 | WWOX | 16q23.1 FRA16D | 1,113,255 | WW domain containing oxidoreductase, putative oxidoreductase |
| 27 | SUCLG2 | 3p14.1 | 294,155 | Succinate-CoA ligase GDP-forming beta subunit, mitochondrial, TCA cycle |
| 28 | KIAA1328 | 18q12.2 FRA18A[a] | 418,211 | Hinderin, competes with SMC1 for binding to SMC3 |
| 29 | HDAC9 | 7p21.1 | 915,475 | Histone dacetylase 9 |
| 30 | TBC1D22A | 22q13.31 FRA22A?[a] | 440,863 | TBC1 domain family member 22A, putative GTPase-activating protein for Rab family protein(s) |
| 31 | TRAPPC9 | 8q24.3 FRA8D | 730,499 | Trafficking protein particle complex 9, vesicular transport from endoplasmic reticulum to Golgi |
| 32 | DIAPH2 | Xq21.33 FRAXL | 920,335 | Diaphanous related formin 2, endosome dynamics |

[a]Cytogenetically characterized by Debacker and Frank Kooy (2007).
[b]Cytogenetically characterized folate sensitive FS at 16p12.3 (Sutherland 1988).

of mitochondrial UPR response element 1 (MURE1) and MURE2 sequence motifs[46,47] both in the untreated condition and following APH treatment (Fig. 3b). We also identified recurrent sequences of 54 or 63 bp containing combined MURE1, CHOP, and MURE2 elements at the promoter and/or in the body of some CFS genes (Fig. 3a and Supplementary Fig. 2). This triplet of elements form a functional module required for mtUPR regulation[47]. Together, these data suggest that mitochondrial stress signaling and FANCD2 play a role in the transcription and stability of CFS genes.

**CFS transcription and breakage is associated with OXPHOS dysfunction.** Mitochondria are a major source of reactive oxygen species that can activate the mtUPR by damaging mitochondrial proteins and decreasing mitochondrial import efficiency[48,49]. Since FANCD2 interacts with MURE elements, we evaluated the OXPHOS activity in FANCD2-deficient cells. Downregulation of FANCD2 elicited a specific defect in electron transport between complexes I and III of the respiratory chain, associated to an uncoupled status, leading to increased oxygen consumption and decreases in ATP synthesis and the ATP/AMP ratio (Fig. 4a). Impaired mitochondrial energy production and a decreased ATP/AMP ratio were accompanied by increased lactate dehydrogenase (LDH) activity at time points after FANCD2 depletion, suggesting a shift to glycolytic metabolism to compensate for the OXPHOS defect (Supplementary Fig. 3a–c). To test whether CFS gene expression is dependent on mitochondrial OXPHOS activity, we treated cells with sodium azide ($NaN_3$) to inhibit mitochondrial respiration and observed decreased expression of all tested CFS genes in both control and FANCD2-depleted cells (Fig. 4b).

To ascertain whether physiological attenuation of OXPHOS metabolism attenuates CFS gene expression and instability, we cultured cells at low oxygen tension (3%) and measured CFS transcription. Compared to cells cultured in 20% $O_2$, FANCD2-depleted cells cultured in 3% $O_2$ showed a significant reduction in the transcription of CFS genes, except for *PARK2*, which was upregulated under low oxygen concentration (Fig. 4c); *PARK2* expression may be induced to promote the shift to glycolytic or fatty acid metabolism[50–52]. We further observed that the global frequency of breaks in metaphase chromosomes at 3% $O_2$ after

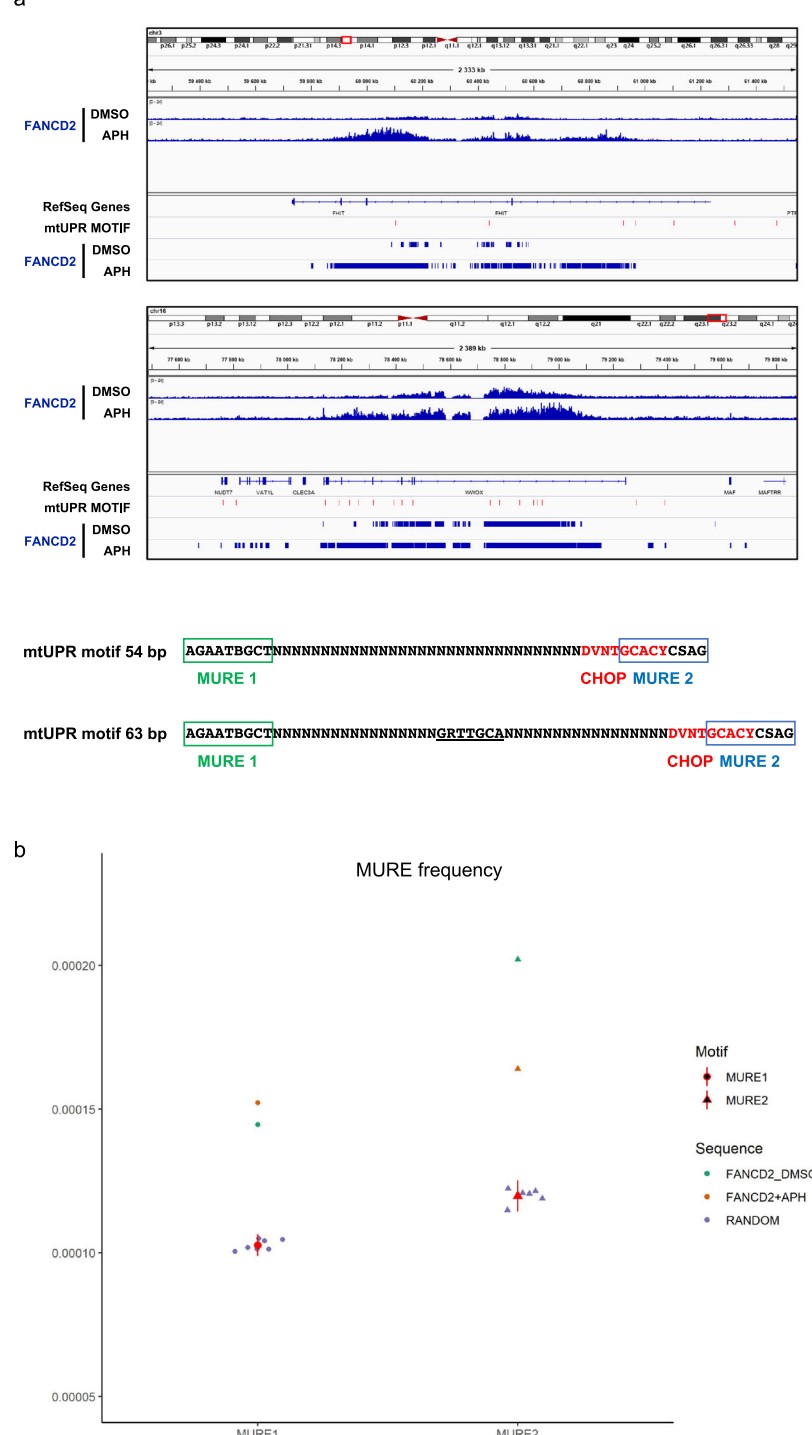

**Fig. 3 Genome-wide analysis of FANCD2 targets identifies an enrichment for mitochondrial UPR-response elements at FANCD2-binding sites. a** IGV visualization of FANCD2 enrichment along the CFS genes *FHIT* and *WWOX* in the presence or absence of APH and the relative position of mitochondrial UPR (mtUPR) motifs (red tags) and FANCD2 peaks. FANCD2 ChIP seq data were scanned with regulatory sequence analysis tools (RSAT)-matrix-scan[89,90] to identify instances of MURE1, MURE2 (mitochondrial UPR response elements) and CHOP motifs as they were defined by Aldridge et al. (2007) and Munch and Harper (2016)[46,47]. mtUPR motifs represent sequences of 54 or 63 bp with MURE1–CHOP–MURE2 consensus elements as indicated. The conserved CHOP consensus between MURE1 and MURE2 elements as reported by Aldridge et al. is underlined. Notice that a 10 bp consensus sequence for CHOP described in Munch and Harper is partially overlapping the MURE2 element. **b** Frequency of MURE1 and MURE2 elements at FANCD2-binding sites relative to a random control. The matches scored for each motif were compared to those detected in $n = 7$ (MURE1) or $n = 6$ (MURE2) independent sets of random sequences of identical length from human genome GRCh37-hg19. $p$ values of the probability to obtain a similar score were calculated with the chi-square test. ****$p = 5.84e-47$ (MURE1), ****$p = 1.91e-174$ (MURE2), and ****$p = 1.85e-69$ (MURE1), ****$p = 9.30e-70$ (MURE2) for FANCD2 in mock (DMSO) and APH-treated conditions, respectively.

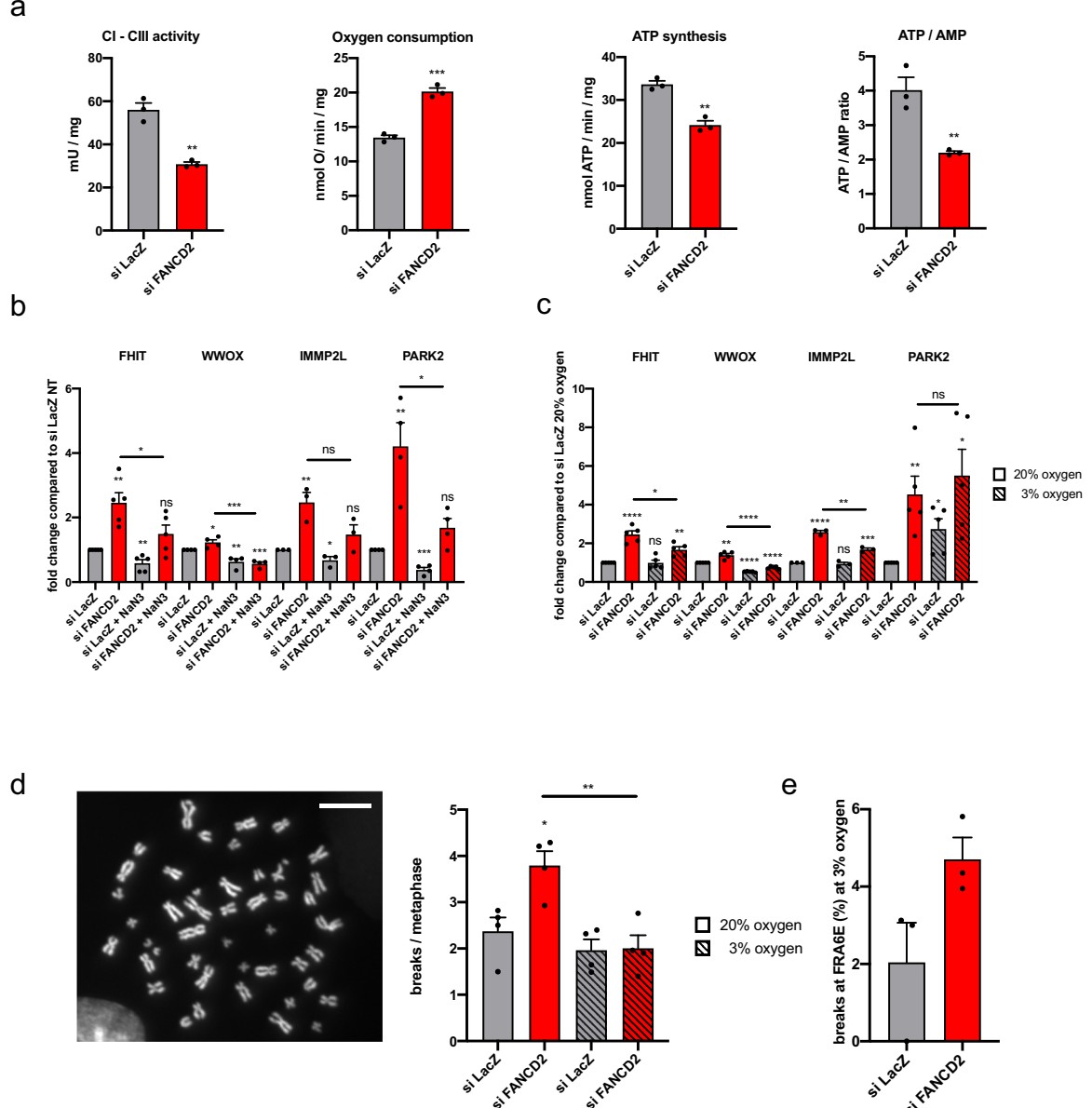

**Fig. 4 CFS expression and breakage is associated with OXPHOS dysfunction. a** Parameters for mitochondrial activity and cellular energy status analyzed after control or FANCD2 siRNA transfection. $n = 3$ independent experiments. **$p = 0.0015$ (CI–CIII activity); ***$p = 0.0004$ (oxygen consumption); **$p = 0.0017$ (ATP synthesis); **$p = 0.008$ (ATP/AMP ratio). **b** RT-qPCR-based analysis of CFS gene expression in cells after control or FANCD2 siRNA transfection and treated or not with 20 mM NaN3 for 10 h. $n = 5$ (FHIT), $n = 4$ (WWOX), $n = 3$ (IMMP2L), $n = 4$ (PARK2) independent experiments. **$p = 0.0017$ (FHIT_siFANCD2), **$p = 0.0052$ (FHIT_siLacZ_NaN3), *$p = 0.0491$ (FHIT_siFANCD2_NaN3 vs. FHIT_siFANCD2), *$p = 0.0288$ (WWOX_siFANCD2), **$p = 0.006$ (WWOX_siLacZ_NaN3), ***$p = 0.0002$ (WWOX_siFANCD2_NaN3), ***$p = 0.0005$ (WWOX_siFANCD2_NaN3 vs. siFANCD2), **$p = 0.0091$ (IMMP2L_siFANCD2), *$p = 0.0464$ (IMMP2L_siLacZ_NaN3), **$p = 0.0047$ (PARK2_siFANCD2), ***$p = 0.0002$ (PARK2_siLacZ_NaN3), *$p = 0.0184$ (PARK2_siFANCD2_NaN3 vs. PARK2_siFANCD2). **c** RT-qPCR analysis of CFS gene expression after control or FANCD2 siRNA transfection of cells maintained at 20% or 3% oxygen. $n = 5$ (FHIT), $n = 5$ (WWOX), $n = 3$ (IMMP2L), $n = 5$ (PARK2) independent experiments. ****$p < 0.0001$ (FHIT_siFANCD2_20%), **$p = 0.0019$ (FHIT_siFANCD2_3%), *$p = 0.011$ (FHIT_siFANCD2_3% vs. FHIT_siFANCD2_20%), **$p = 0.0013$ (WWOX_siFANCD2_20%), ****$p < 0.0001$ (WWOX_siLacZ_3%), ****$p < 0.0001$ (WWOX_siFANCD2_3%), ****$p < 0.0001$ (WWOX_siFANCD2_3% vs. WWOX_siFANCD2_20%), ****$p < 0.0001$ (IMMP2L_siFANCD2_20%), ***$p = 0.0008$ (IMMP2L_siFANCD2_3%), **$p = 0.0014$ (IMMP2L_siFANCD2_3% vs. IMMP2L_siFANCD2_20%), **$p = 0.006$ (PARK2_siFANCD2_20%), *$p = 0.0118$ (PARK2_siLacZ_3%), *$p = 0.0107$ (PARK2_siFANCD2_3%). **d** Chromosome fragility in cells transfected with control or FANCD2 siRNA, treated with 0.3 μM APH and maintained at 3% or 20% oxygen. Left, example of a DAPI-stained metaphase spread; the arrows indicate a break. Right, total breaks are scored as the mean number of breaks per metaphase. A total of 196 (siLacZ_20%), 221 (siFANCD2 20%), 185 (siLacZ 3%), and 186 (siFANCD2_3%) metaphases were analyzed from $n = 4$ independent experiments. *$p = 0.0166$ (siFANCD2_20%), **$p = 0.0053$ (siFANCD2_3% vs. siFANCD2_20%). **e** Frequency of FRA6E breaks in cells maintained at 3% oxygen, transfected with control or FANCD2 siRNA and treated with 0.3 μM APH. A total of 119 (siLacZ_3%), and 104 (siFANCD2_3%) metaphases were analyzed from $n = 3$ independent experiments. Error bars are standard error of the mean (SEM).

depletion of FANCD2 was significantly reduced compared with cells cultured in 20% $O_2$ (Fig. 4d), even if a low frequency of breaks specifically occurred at *PARK2*/FRA6E (Fig. 4e), revealing a close correlation between the level of CFS gene transcription and instability. Taken together, these data indicate that CFS gene expression is linked to mitochondrial activity and that chromosome fragility in FANCD2-depleted cells is associated with mitochondrial dysfunction.

**Mitochondrial stress triggers CFS gene expression and FANCD2 binding at CFSs.** We then sought to investigate the role of mitochondrial stress signaling in CFS transcription and FANCD2 function. Mitochondrial stress can be triggered by protein folding stress in the mitochondria, which communicate with the ER via mitochondria-associated ER membrane (MAM) contacts[53,54]. To determine if stress signaling originating in either the mitochondria or the ER is involved in CFS gene transcription, we pharmacologically induced the UPR in the respective organelles using carbonyl cyanide m-chlorophenyl hydrazone (CCCP), a mitochondrial uncoupler, or thapsigargin (TG), a sarco/ER $Ca^{2+}$-ATPase (SERCA) inhibitor and inducer of ER stress. Both treatments induced CFS gene transcription, which was further increased after FANCD2 depletion, indicating that CFS genes respond to mitochondrial or ER stress-dependent UPR activation and that FANCD2 dampens this response (Fig. 5a). Treatment with TG or CCCP induced FANCD2 relocalization into nuclear foci, some of which persisted in mitosis, as observed after APH treatment, suggesting that they correspond to CFSs (Fig. 5b, c). We performed FANCD2 ChIP followed by qPCR after TG or CCCP treatment and detected enrichment of FANCD2 at CFS genes (Fig. 5d), demonstrating that mitochondrial or ER-stress-dependent UPR signaling promotes the recruitment of FANCD2 to CFSs. Similarly, FANCI was also enriched at CFS genes (Supplementary Fig. 4a). However, TG or CCCP treatments did not induce full activation of the DNA damage response, as assessed by phosphorylation of ATR targets chk1 (pS345) and RPA2 (pS33), and gammaH2AX foci formation, suggesting only minor perturbation of DNA replication[55,56] (Supplementary Fig. 4b).

To selectively perturb mitochondrial proteostasis, we downregulated spastic paraplegia 7 (*SPG7*), which encodes the paraplegin matrix AAA peptidase subunit, a mitochondrially localized membrane-associated protease. SPG7 downregulation increases the load of unfolded proteins in the mitochondria, activating the mtUPR uncoupled from accumulation of reactive oxygen species[48]. Importantly, this allows to distinguish between a signal directly originating in the mitochondria from secondary effects due to crosstalk between mitochondria and ER in redox regulation and calcium signaling[57]. SPG7 depletion by RNAi increased CFS gene transcription (Fig. 5e), as observed after FANCD2 downregulation, demonstrating that CFS gene expression is triggered by mitochondrial-dependent stress signaling and that FANCD2 counteracts mitochondrial stress. The fact that CFS gene transcription can be induced independently by depletion of FANCD2 or SPG7 suggests that they participate in processes, such as mitochondrial gene expression, protein synthesis and mitophagy[33,34], and mitochondrial protein processing[58], that functionally cooperate in mitochondrial biogenesis and protein quality control[59]. Interestingly, we found a slight but significant induction of *SPG7* transcription in FANCD2-deficient cells (Supplementary Fig. 4c), consistent with its conserved function in mtUPR in mammalian cells[60,61].

**FANCD2 functionally interacts with UBL5 in mtUPR signaling and CFS stability.** To understand which pathway is involved

mitochondrial stress signaling and regulation of CFS genes, we examined mtUPR markers and factors involved in mitochondrial stress signaling. ATF4 is a transcription factor activated upon UPR induction and a key effector of the mitochondrial stress response in mammalian cells[62]. FANCD2 downregulation induced ATF4 transcription, its nuclear relocalization, and activation of its targets *CHAC1*, *PCK2*, and *PSAT1* (Fig. 6a, b), indicating the activation of mitochondrial stress signaling. ATF4 knockdown decreased the expression of these genes and abrogated their induction after FANCD2 depletion (Fig. 6b). However, ATF4 depletion did not restore the increased CFS gene transcription observed after FANCD2 depletion (Supplementary Fig. 5a), suggesting that CFS genes may be regulated by FANCD2 in an ATF4-independent manner. To support this finding, we examined the role of the integrated stress response (ISR), which can be induced by both mitochondrial and ER stress and is involved in the ATF4-dependent mitochondrial stress signaling[62,63]. We found that ISR inhibition by ISRIB modestly reduced the TG-incited or CCCP-incited expression of CFS genes but did not prevent their induction in FANCD2-deficient cells, further suggesting that the CFS genes can be induced by a separate pathway (Fig. 6c, d).

We then analyzed the expression of the mitochondrial and/or ER-specific UPR markers CCAAT/enhancer-binding protein homology protein (*CHOP*)[64] and binding-immunoglobulin protein (*BiP*)[65,66], respectively. Under untreated conditions, *CHOP* or *BiP* expression was not affected by FANCD2. However, FANCD2 downregulation significantly impaired the induction of *CHOP* after CCCP and less so after TG, while it did not significantly affect *BIP*, confirming the role of FANCD2 in regulating the mtUPR (Fig. 6e, f).

Treatment with ISRIB also reduced *CHOP* expression, and had an additive effect with FANCD2 depletion in decreasing *CHOP* and *BiP* expression (Fig. 6e, f), suggesting that FANCD2 and the ISR may operate in parallel pathways in response to mitochondrial and ER stress. Mitochondrial dysfunction elicited by defects in the mitochondrial helicase Twinkle activates a multifaceted tissue-specific integrated mitochondrial stress response (ISRmt), which is regulated by mTORC1, and comprises both the ATF4 branch of the mitochondrial stress response and the mtUPR[67].

Interfering with the expression of the mitochondrial genome, which leads to accumulation of unassembled subunits of mitochondrial complexes, or overexpression of a folding impaired matrix enzyme, selectively induces the expression of mitochondrial chaperones, notably the chaperonins Hsp60 (Hspd1) and Hsp10 (Hspe1), mtDNAJ, and the protease ClpP in mammalian cells[64,68]. Hematopoietic cells from *Fancd2* knock-out mice also show mito-nuclear protein imbalance and increased protein levels of Hsp60/Hsp10, mtDnaJ and ClpP[33]. We observed a slight increase of HSP60 protein levels after FANCD2 depletion in our cells (Supplementary Fig. 5b), but did not find a significant induction of *HSPD1* transcript levels (Supplementary Fig. 5c). This may be due to cell-type and tissue-specific differences, and to the unfolded protein level-dependent and time-dependent nature of chaperonin induction[61,64]. Noticeably, we found that FANCD2 was bound to the common bidirectional promoter of *HSPD1* and *HSPE1* genes (Supplementary Fig. 5d), suggesting that FANCD2 itself may regulate their expression.

UBL5 is a ubiquitin-like protein involved in the mtUPR in *C. elegans*[69] that has been shown to promote the functional integrity of the FANC pathway[70].

To determine if UBL5 may participate with FANCD2 in the mtUPR, we analyzed *CHOP* and *BiP* induction by CCCP or TG treatments. Strikingly, depleting UBL5 had an effect comparable to FANCD2, and similarly decreased *CHOP* expression after CCCP (Supplementary Fig. 5,e, f).

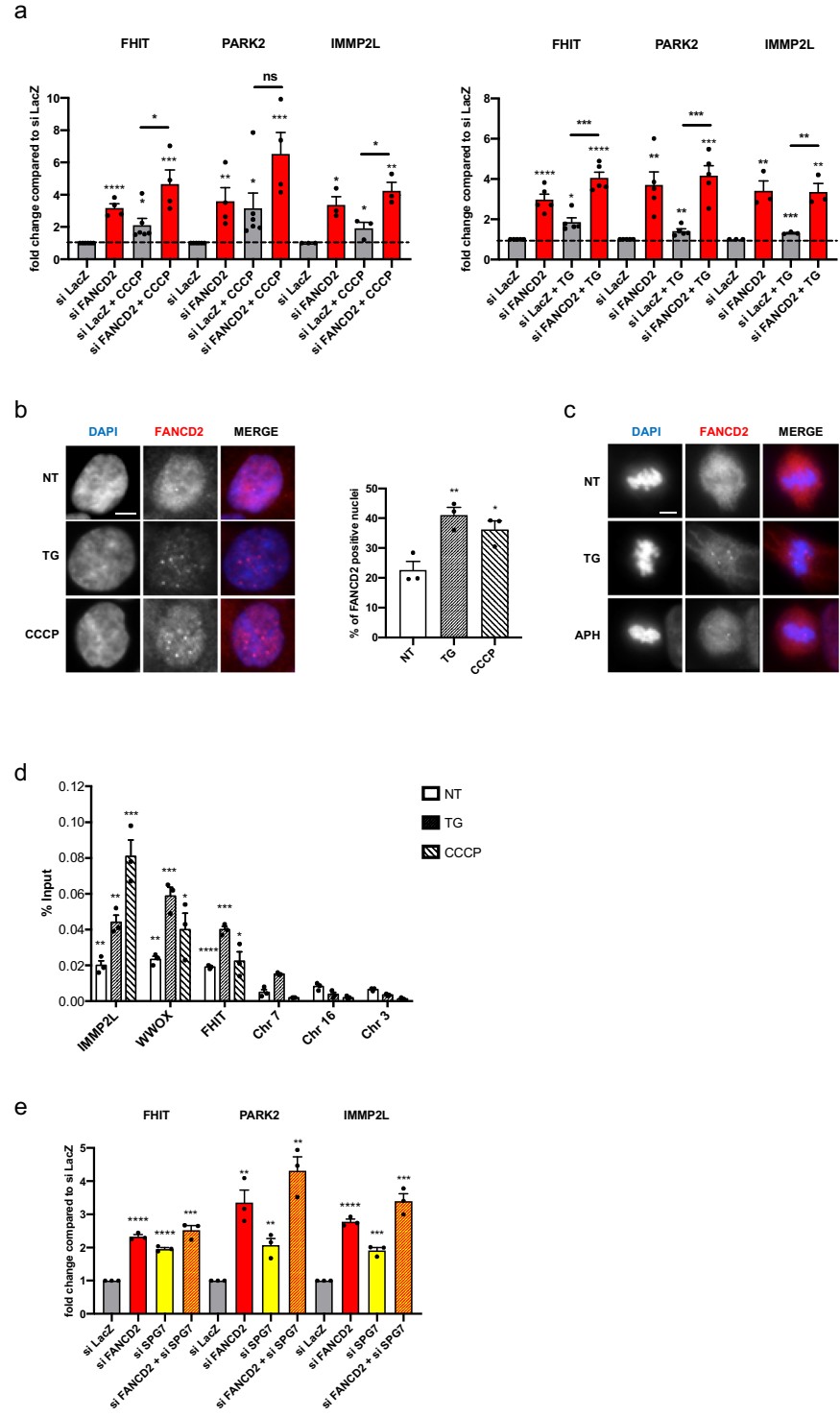

Then, to determine if UBL5 is involved in regulation of CFS genes, we downregulated UBL5 and measured CFS gene expression. UBL5 knockdown significantly reduced the upregulation of CFS genes in FANCD2-depleted cells in untreated conditions (Fig. 6g), and also partially or completely abrogated their induction after CCCP or TG treatments (Supplementary Fig. 5g, h), revealing that FANCD2 and UBL5 may operate in the same pathway to regulate the mtUPR and CFS stability in human cells. Reducing CFS gene induction by downregulating UBL5 in FANCD2-depleted cells decreased chromosome instability observed after APH treatment to levels similar to control values (Fig. 6h), indicating that suppression

of mitochondrial stress signaling reduces chromosome fragility in the absence of a functional FANC pathway.

Collectively, these data reveal a role of FANCD2 as a regulatory component of the mitochondrial stress response involved in mito-nuclear communication, counteracting mitochondrial stress and attuning UBL5-dependent CFS gene transcription to prevent replication stress and CFS instability (Fig. 7).

## Discussion

The coordinated regulation of mitochondrial and nuclear activities is essential for cellular function and metabolic homeostasis.

**Fig. 5 Mitochondrial stress triggers CFS gene expression and FANCD2 binding at CFSs. a** Left, RT-qPCR-based analysis of CFS gene expression in siLacZ- and siFANCD2-transfected cells treated or not with 10 μM CCCP for 8 h. Right, RT-qPCR-based analysis of CFS gene expression in siLacZ- and siFANCD2-transfected cells treated or not with 1 μM TG for 8 h. $n = 6$ (FHIT_siLacZ), $n = 4$ (FHIT_siFANCD2), $n = 6$ (FHIT_siLacZ_CCCP), $n = 4$ (FHIT_siFANCD2_CCCP), $n = 6$ (PARK2_siLacZ), $n = 4$ (PARK2_siFANCD2), $n = 6$ (PARK2_siLacZ_CCCP), $n = 4$ (PARK2_siFANCD2_CCCP), $n = 3$ (IMMP2L) independent experiments for cells treated or not with CCCP; and $n = 5$ (FHIT), $n = 4$ (PARK2), $n = 3$ (IMMP2L) for cells treated or not with TG. ****$p < 0.0001$ (FHIT_siFANCD2), *$p = 0.0207$ (FHIT_siLacZ_CCCP), ***$p = 0.0007$ (FHIT_siFANCD2_CCCP), *$p = 0.0174$ (FHIT_siFANCD2_CCCP vs. FHIT_siLacZ_CCCP), **$p = 0.0048$ (PARK2_siFANCD2), *$p = 0.047$ (PARK2_siLacZ_CCCP), ***$p = 0.0008$ (PARK2_siFANCD2_CCCP), *$p = 0.0104$ (IMMP2L_siFANCD2), **$p = 0.0037$ (IMMP2L_siFANCD2_CCCP), *$p = 0.023$ (IMMP2L_siFANCD2_CCCP vs. IMMP2L_siLacZ_CCCP; ****$p < 0.0001$ (FHIT_siFANCD2), *$p = 0.0038$ (FHIT_siLacZ_TG), ****$p < 0.0001$ (FHIT_siFANCD2_TG), ***$p = 0.0003$ (FHIT_siFANCD2_TG vs. FHIT_siLacZ_TG), **$p = 0.003$ (PARK2_siFANCD2), **$p = 0.0058$ (PARK2_siLacZ_TG), ***$p = 0.0002$ (PARK2_siFANCD2_TG), ***$p = 0.0006$ (PARK2_siFANCD2_TG vs. PARK2_siLacZ_TG), **$p = 0.008$ (IMMP2L_siFANCD2), ***$p = 0.0006$ (IMMP2L_siLacZ_TG), **$p = 0.0052$ (IMMP2L_siFANCD2_TG), **$p = 0.0087$ (IMMP2L_siFANCD2_TG vs. IMMP2L_siLacZ_TG). **b** Left, examples of immunofluorescence staining of FANCD2 in cells treated with TG or CCCP or in cells that were not treated (NT); FANCD2 is shown in red, and DNA (DAPI) is in blue in the merged image. Right, percentage of FANCD2-positive nuclei; FANCD2 foci were counted in a total of 335 (NT), 318 (TG), and 330 (CCCP) nuclei from $n = 3$ independent experiments, and nuclei with more than five spots were quantified. Scale bars, 5 μM. **c** Immunofluorescence staining of FANCD2 in metaphase cells treated or NT with TG or APH. Scale bars, 5 μM. **d** FANCD2 ChIP followed by qPCR in cells treated or not with 1 μM TG or 10 μM CCCP for 8 h. The results are expressed as the percentage of the input. Chromosome 7 (Chr 7), Chromosome 16 (Chr 16), and Chromosome 3 (Chr 3) were used as control regions close to the *IMMP2L*, *WWOX*, and *FHIT* genes, respectively. $n = 3$ independent experiments. **$p = 0.0084$ (IMMP2L_NT), **$p = 0.002$ (IMMP2L_TG), ***$p = 0.001$ (IMMP2L_CCCP), **$p = 0.002$ (WWOX_NT), **$p = 0.0004$ (WWOX_TG), *$p = 0.139$ (WWOX_CCCP), ****$p < 0.0001$ (FHIT_NT), ****$p < 0.0001$ (FHIT_TG), *$p = 0.161$ (FHIT_CCCP). **e** Expression of large CFS genes measured by RT-qPCR after control, FANCD2, SPG7, or FANCD2 and SPG7 siRNA transfection. $n = 3$ independent experiments. ****$p < 0.0001$ (FHIT_siFANCD2), ****$p < 0.0001$ (FHIT_siSPG7), ***$p = 0.0004$ (FHIT_siFANCD2 + siSPG7), **$p = 0.0036$ (PARK2_siFANCD2), **$p = 0.0066$ (PARK2_siSPG7), **$p = 0.0014$ (PARK2_siFANCD2 + siSPG7), ****$p = 0.0001$ (IMMP2L_siFANCD2), ***$p = 0.0006$ (IMMP2L_siSPG7), ***$p = 0.0005$ (IMMP2L_siFANCD2 + siSPG7). Error bars are standard error of the mean (SEM).

Mitochondria are signaling hubs whose activity during oxidative metabolism is monitored through the levels of metabolites, nucleotides, reactive oxygen species, the rate of ATP production, or the level of misfolded proteins[71]. Retrograde pathways, called UPRs, signal organelle-specific and compartment-specific stress and activate nuclear programs to adjust cellular metabolic output to regain homeostasis. In this study, we show that FANCD2 participates in the mitochondrial UPR pathway, and mitochondrial nuclear crosstalk, in human cells.

FANC proteins function in the maintenance of genome stability. However, they also perform non-canonical functions in mitochondria[32,72]. Here, we reveal a dual function for FANCD2 in counteracting mitochondrial stress and in dampening the mtUPR-induced transcription of CFS genes. Interestingly, a similar role has been reported for the *C elegans* respiratory enzyme clk-1 and its human homolog COQ7[60]. Large CFS genes may be exquisitely sensitive rheostats of cellular metabolic activity. For example, variations in dNTP biosynthesis and ROS directly modulate replisome architecture and replication fork velocity[73]. CFSs are late replicating, and slowing their replication increases the risk of incomplete replication and chromosome breakage at mitosis[9]. Furthermore, the timing of CFS replication is modulated by transcription[7], and the failure to coordinate these two processes leads to CFS breakage[4].

We identify a regulatory mechanism that links CFS transcription to mitochondrial activity. Depletion of FANCD2 induces a mitochondrial stress response that resembles the ISRmt and activates the ATF4 pathway, which rewires mitochondrial metabolism[62,67,74,75] and may be beneficial, at least on a short term, during the recovery from stress. We show that a distinct branch of this response, which functions independently of ATF4, regulates CFS transcription and is modulated by a UBL5–FANCD2 axis. In this context, UBL5 acts as an activator and FANCD2 as a suppressor, which may constitute a feedback loop for fine tuning the mtUPR. Indeed, it has been reported that UBL5 stabilizes FANCI and FANCD2 and promotes their interaction[70], which may titrate UBL5, leading to mtUPR attenuation. It is likely that transient activation of CFS genes is also required to recover mitochondrial or ER homeostasis, as exemplified by *PARK2/Parkin* expression[76]. FANC

proteins not only interact with Parkin[34] to promote mitophagy but they also tune its expression (Figs. 1a, 4a, e, 5b, c, i). In this scenario, mild mitochondrial stress would be beneficial and constitute a feedback mechanism to ensure cellular homeostasis. However, impaired mito-nuclear communication would lead on the one side to prolonged or excessive mitochondrial dysfunction, and activation of the mitochondrial stress response, on the other, to replication stress and conflicts between replication and transcription (Fig. 6). Indeed, it has been reported that TG-mediated UPR induction reduces replication fork progression and origin firing[56]. We did not detect a general induction of the nuclear DNA damage response at the doses of TG or CCCP we used in our study. It is likely that the effects of transcription on replication dynamics are dose dependent[7] and may differently affect fork progression and origin positioning, thus resetting the cell replication program[77].

We demonstrate that transcription is required for FANCD2 recruitment to CFS genes and for its function in CFS maintenance. In addition, we find that FANCD2 enrichment at CFSs is proportional to the level of CFS gene transcription (Supplementary Fig. 1b) and is promoted both by mitochondrial stress induction (Fig. 5c, d) and replication stress (Fig. 3b and Supplementary Fig. 2). Therefore, we propose that CFS loci behave as both cis-acting and trans-acting components of the mitochondrial stress response that become unstable above a threshold of transcriptional activation and replication stress. The encounters between replication and transcription and R-loop formation would generate the substrate for FANCD2 binding and/or retention at CFSs and trigger the activation of the FANC pathway[22,27,28,78], constituting a metabolic and genome surveillance checkpoint. Indeed, FANCD2 and FANCI dynamically interact with components of the transcription machinery[79] and are recruited to stalled replication forks[80]. Recently, it has been shown that interaction between SLX4/FANCP and the DNA helicase RTEL1 drives the assembly of FANCD2 foci in the vicinity of RNA polymerase II[81], to prevent endogenous transcription-induced replication stress. It would be interesting to test whether FANCD2/FANCI–UBL5 is involved in this pathway. We propose that a FANCD2/FANCI–UBL5 axis is an integral

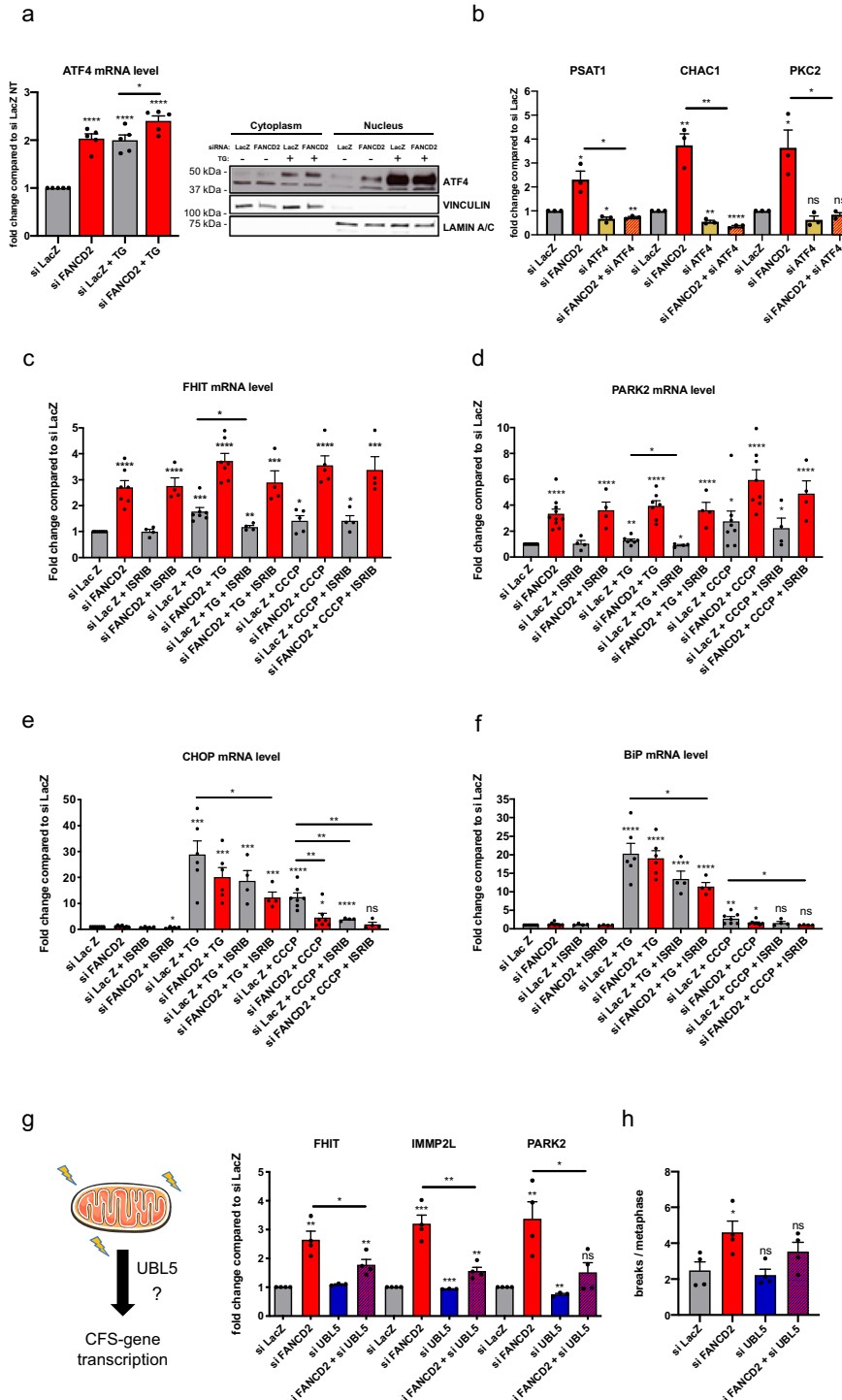

component of a signaling cascade that allows the "ground control" to get information on mitochondria's status and to respond to metabolic perturbations[82].

In *C. elegans*, the ubiquitin-like protein UBL-5 regulates the mtUPR in parallel to the transcription factor ATFS-1[69]. Upon mitochondrial stress, UBL-5 regulates gene expression by interacting with homeodomain-containing transcription factor DVE-1, which correlates with temporal and spatial redistribution of DVE-1 in nuclear puncta and its enhanced binding to promoters of mitochondrial chaperone genes[83]. Sequence analysis has identified the chromatin organizers AT-rich sequence-binding SATB1 and SATB2 as the closest mammalian homologs of DVE-

1, though their functional implication in the mtUPR is unclear[83]. Our findings suggest that FANCD2 and FANCI may perform in complex with UBL5 a similar function in modulating the mtUPR.

We find that FANCD2-binding sites are enriched in MURE elements, and that CFS genes contain a variable number of mtUPR modules of 54 or 63 bp (Fig. 3a and Supplementary Fig. 2). As both MURE1 and MURE2 and the combination of MURE1–CHOP–MURE2 motifs are involved in mtUPR regulation[46,47], we speculate that copy number variations (CNVs) and rearrangements targeting CFSs may affect the magnitude of the CFS gene response to mtUPR and, as a consequence, cellular stress resistance and metabolic adaptation[84] during normal

**Fig. 6 FANCD2 functionally interacts with UBL5 in mitochondrial UPR signaling and CFS stability. a** ATF4 expression measured by RT-qPCR in siLacZ- and siFANCD2-transfected cells, treated or not with 1 μM TG (left panel). $n = 5$ independent experiments. ****$p < 0.0001$. Western Blot detection of ATF4 in cytoplasmic and nuclear fractions of siLacZ- or siFANCD2-transfected cells treated (+) or not (−) with 1 μM TG (right panel). Vinculin and lamin A/C were used as loading controls for cytoplasmic and nuclear fractions, respectively. **b** Expression of ATF4 target genes measured by RT-qPCR after control, FANCD2, ATF4, or FANCD2 and ATF4 siRNA transfection. $n = 3$ independent experiments. *$p = 0.0194$ (PSAT1_siFANCD2), *$p = 0.0134$ (PSAT1_siATF4), **$p = 0.0015$ (PSAT1_siFANCD2 + siATF4), *$p = 0.0105$ (PSAT1_siFANCD2 + siATF4 vs. PSAT1_siFANCD2); **$p = 0.0048$ (CHAC1_siFANCD2), **$p = 0.015$ (CHAC1_siATF4), ****$p < 0.0001$ (CHAC1_siFANCD2 + siATF4), **$p = 0.0022$ (CHAC1_siFANCD2 + siATF4 vs. CHAC1_siFANCD2); *$p = 0.0239$ (PKC2_siFANCD2), *$p = 0.0201$ (PKC2_siFANCD2 + siATF4 vs. PKC2_siFANCD2). **c** FHIT mRNA level measured by RT-qPCR in siLacZ- or siFANCD2-transfected cells treated or not with 1 μM TG or 10 μM CCCP and ISRIB (500 nM) for 8 h. $n = 7$ (siLacZ, siFANCD2, siLacZ + TG, siFANCD2 + TG), $n = 5$ (siLacZ + CCCP, siFANCD2 + CCCP), $n = 4$ (siLacZ + ISRIB, siFANCD2 + ISRIB, siLacZ + TG + ISRIB, siFANCD2 + TG + ISRIB, siLacZ + CCCP + ISRIB, siFANCD2 + CCCP + ISRIB) independent experiments. ****$p < 0.0001$ (siFANCD2), ****$p < 0.0001$ (siFANCD2 + ISRIB), ***$p = 0.0004$ (siLacZ + TG), ****$p < 0.0001$ (siFANCD2 + TG), **$p = 0.0047$ (siLacZ + TG + ISRIB), *$p = 0.0241$ (siLacZ + TG + ISRIB vs. siLacZ + TG), ***$p = 0.0003$ (siFANCD2 + TG + ISRIB), *$p = 0.026$ (siLacZ + CCCP), ****$p < 0.0001$ (siFANCD2 + CCCP), *$p = 0.0202$ (siLacZ + CCCP + ISRIB), ***$p = 0.0001$ (siFANCD2 + CCCP + ISRIB). **d** PARK2 mRNA level measured by RT-qPCR in siLacZ-transfected or siFANCD2-transfected cells treated or not with 1 μM TG or 10 μM CCCP and ISRIB (500 nM) for 8 h. $n = 10$ (siLacZ, siFANCD2), $n = 8$ (siLacZ + CCCP, siFANCD2 + CCCP), $n = 7$ (siLacZ + TG, siFANCD2 + TG), $n = 4$ (siLacZ + ISRIB, siFANCD2 + ISRIB, siLacZ + TG + ISRIB, siFANCD2 + TG + ISRIB, siLacZ + CCCP + ISRIB, siFANCD2 + CCCP + ISRIB) independent experiments. ****$p < 0.0001$ (siFANCD2), ****$p < 0.0001$ (siFANCD2 + ISRIB), **$p = 0.0045$ (siLacZ + TG), ****$p < 0.0001$ (siFANCD2 + TG), *$p = 0.0312$ (siLacZ + TG + ISRIB), *$p = 0.0328$ (siLacZ + TG + ISRIB vs. siLacZ + TG), ****$p < 0.0001$ (siFANCD2 + TG + ISRIB), *$p = 0.022$ (siLacZ + CCCP), ****$p < 0.0001$ (siFANCD2 + CCCP), *$p = 0.0203$ (siLacZ + CCCP + ISRIB), ****$p < 0.0001$ (siFANCD2 + CCCP + ISRIB). **e** CHOP mRNA level measured by RT-qPCR in siLacZ-transfected or siFANCD2-transfected cells treated or not with 1 μM TG or 10 μM CCCP and ISRIB (500 nM) for 8 h. $n = 7$ (siLacZ, siFANCD2, siLacZ + CCCP, siFANCD2 + CCCP), $n = 6$ (siLacZ + TG, siFANCD2 + TG), $n = 4$ (siLacZ + ISRIB, siFANCD2 + ISRIB, siLacZ + TG + ISRIB, siFANCD2 + TG + ISRIB, siLacZ + CCCP + ISRIB, siFANCD2 + CCCP + ISRIB) independent experiments. *$p = 0.0221$ (siFANCD2 + ISRIB), ***$p = 0.0001$ (siLacZ + TG), ***$p = 0.0001$ (siFANCD2 + TG), ***$p = 0.0002$ (siLacZ + TG + ISRIB), ****$p < 0.0001$ (siFANCD2 + TG + ISRIB), *$p = 0.0382$ (siFANCD2 + TG + ISRIB vs. siLacZ + TG), ****$p < 0.0001$ (siLacZ + CCCP), *$p = 0.0477$ (siFANCD2 + CCCP), ****$p < 0.0001$ (siLacZ + CCCP + ISRIB), **$p = 0.0046$ (siFANCD2 + CCCP vs. siLacZ + CCCP), **$p = 0.0033$ (siLacZ + CCCP + ISRIB vs. siLacZ + CCCP), **$p = 0.0033$ (siFANCD2 + CCCP + ISRIB vs. siLacZ + CCCP). **f** BIP mRNA level measured by RT-qPCR in siLacZ-transfected or siFANCD2-transfected cells treated or not with 1 μM TG or 10 μM CCCP and ISRIB (500 nM) for 8 h. $n = 7$ (siLacZ, siFANCD2, siLacZ + CCCP, siFANCD2 + CCCP), $n = 6$ (siLacZ + TG, siFANCD2 + TG), $n = 4$ (siLacZ + ISRIB, siFANCD2 + ISRIB, siLacZ + TG + ISRIB, siFANCD2 + TG + ISRIB, siLacZ + CCCP + ISRIB, siFANCD2 + CCCP + ISRIB) independent experiments. ****$p < 0.0001$ (siLacZ + TG), ****$p < 0.0001$ (siFANCD2 + TG), ****$p < 0.0001$ (siLacZ + TG + ISRIB), ****$p < 0.0001$ (siFANCD2 + TG + ISRIB), **$p = 0.008$ (siLacZ + CCCP), *$p = 0.0244$ (siFANCD2 + CCCP), *$p = 0.0361$ (siFANCD2 + TG + ISRIB vs. siLacZ + TG), *$p = 0.0467$ (siFANCD2 + CCCP + ISRIB vs. siLacZ + CCCP). **g** Left, graphical representation of mitochondrial UPR activation of CFS genes. Right, expression of large CFS genes measured by RT-qPCR after control, FANCD2, UBL5, or FANCD2 and UBL5 siRNA transfection. $n = 4$ (siLacZ, siFANCD2, siFANCD2 + siUBL5), $n = 3$ (siUBL5) independent experiments. **$p = 0.014$ (FHIT_siFANCD2), **$p = 0.0062$ (FHIT_siFANCD2 + siUBL5), *$p = 0.0461$ (FHIT_siFANCD2 + siUBL5 vs. FHIT_siFANCD2); ***$p = 0.0003$ (IMMP2L_siFANCD2), ****$p < 0.0001$ (IMMP2L_siUBL5), **$p = 0.0062$ (IMMP2L_siFANCD2 + siUBL5), **$p = 0.0025$ (IMMP2L_siFANCD2 + siUBL5 vs. IMMP2L_siFANCD2); **$p = 0.0067$ (PARK2_siFANCD2), ***$p = 0.0004$ (PARK2_siUBL5), *$p = 0.0328$ (PARK2_siFANCD2 + siUBL5 vs. PARK2_siFANCD2). **h** Chromosome fragility in cells transfected with control, FANCD2, UBL5, or FANCD2 and UBL5 siRNA and treated with 0.3 μM APH, scored as the mean number of breaks per metaphase. A total of 202 (siLacZ), 202 (siFANCD2), 194 (siUBL5), and 202 (siFANCD2 + siUBL5) metaphases were analyzed from $n = 4$ independent experiments. *$p = 0.035$ (siFANCD2). Error bars are standard error of the mean (SEM).

development and in cancer[85]. The mitochondrial and ER UPR pathways are major metabolic checkpoints that regulate hematopoietic stem cell function and integrity[86–88]. It will be crucial to characterize how the FA pathway regulates the mitochondrial UPR during hematopoiesis and whether it modulates the expression and stability of specific CFSs in hematopoietic cells.

## Methods

**Cell culture.** The HCT116 cell line was maintained in McCoy's 5A medium (ATCC), and RKO and HeLa cells were maintained in Dulbecco's modified Eagle's medium (DMEM) (Gibco) at 37 °C in a humidified atmosphere under 5% $CO_2$. Media were supplemented with 10% fetal bovine serum (FBS), 1 mM sodium pyruvate, 100 U/ml penicillin and 100 μg/ml streptomycin. All cell lines were purchased from ATCC. HCT116 FHIT-KO cells were generated by CRISPR/Cas9 genome editing of parental HCT116 cells.

For experiments requiring low oxygen conditions, cells were maintained in an incubator with an $O_2$ control system (HERAcell 150i, Thermo Scientific).

**siRNA transfection.** siRNA duplex oligonucleotides were purchased from Ambion to target FHIT (#AM16708) and from Eurogentec to target the other assayed genes. The siRNA sequences are provided in Supplementary Table S1. For all siRNA experiments, cells were transfected with siRNAs at a final concentration of 20 nM using INTERFERin (Polyplus) according to the manufacturer's instructions. Following siRNA transfection, knockdown of gene expression was assessed by Western blot or qRT-PCR analysis. Unless otherwise indicated, cells were collected for total cell lysate preparation, subcellular fractionation, biochemical assays, and qRT-PCR analysis 48 h after transfection.

**Western blotting and subcellular fractionation.** For total lysates, cells were disrupted in lysis buffer (50 mM Tris–HCl, 20 mM NaCl, 1 mM $MgCl_2$, and 0.1% SDS) containing a protease and phosphatase inhibitor cocktail (Roche) supplemented with 0.1% endonuclease (benzonase, Millipore) for 10 min at room temperature with rotation. For fractionation analysis, cells were lysed using an NE-PER kit (ThermoFisher) following the manufacturer's instructions. Laemmli buffer containing beta-mercaptoethanol was added to the samples, which were subsequently boiled for 5 min at 95 °C. The proteins were separated on SDS–PAGE denaturing gels (Bio-Rad) and transferred to nitrocellulose membranes (Bio-Rad). Next, the membranes were blocked with phosphate-buffered saline (PBS)–milk (5%) or PBS–bovine serum albumin (BSA) (3%) for 1 h, and signals were visualized using WesternBright ECL (Advansta) on a digital imaging system (GeneGnome, Syngene) or using Amersham Hyperfilm ECL film (GE) on a table-top processor (Curix 60, AGFA). The antibodies used in this study are listed in Supplementary Table S1.

**Quantitative RT-PCR.** Total cellular RNA was extracted with the ReliaPrep RNA Cell Miniprep System (Promega), and 1 μg of RNA was used to synthesize cDNA with a high-capacity RNA-to-cDNA kit (ThermoFisher). PCR primers were purchased from Eurogentec and used in PCRs with SYBR Green Master Mix (ThermoFisher) on a QuantStudio 7 Flex instrument (Applied Biosystems). Relative gene expression was calculated using the ΔΔCq method and normalized to *GAPDH* expression. Values are represented as the fold change compared to the control transfection values (siLacZ). Primer sequences are available in Supplementary Table S1.

**Cell treatments and chemicals.** Replicative stress was induced by treating cells with 0.3 μM APH (Sigma A0781) for 20 h.

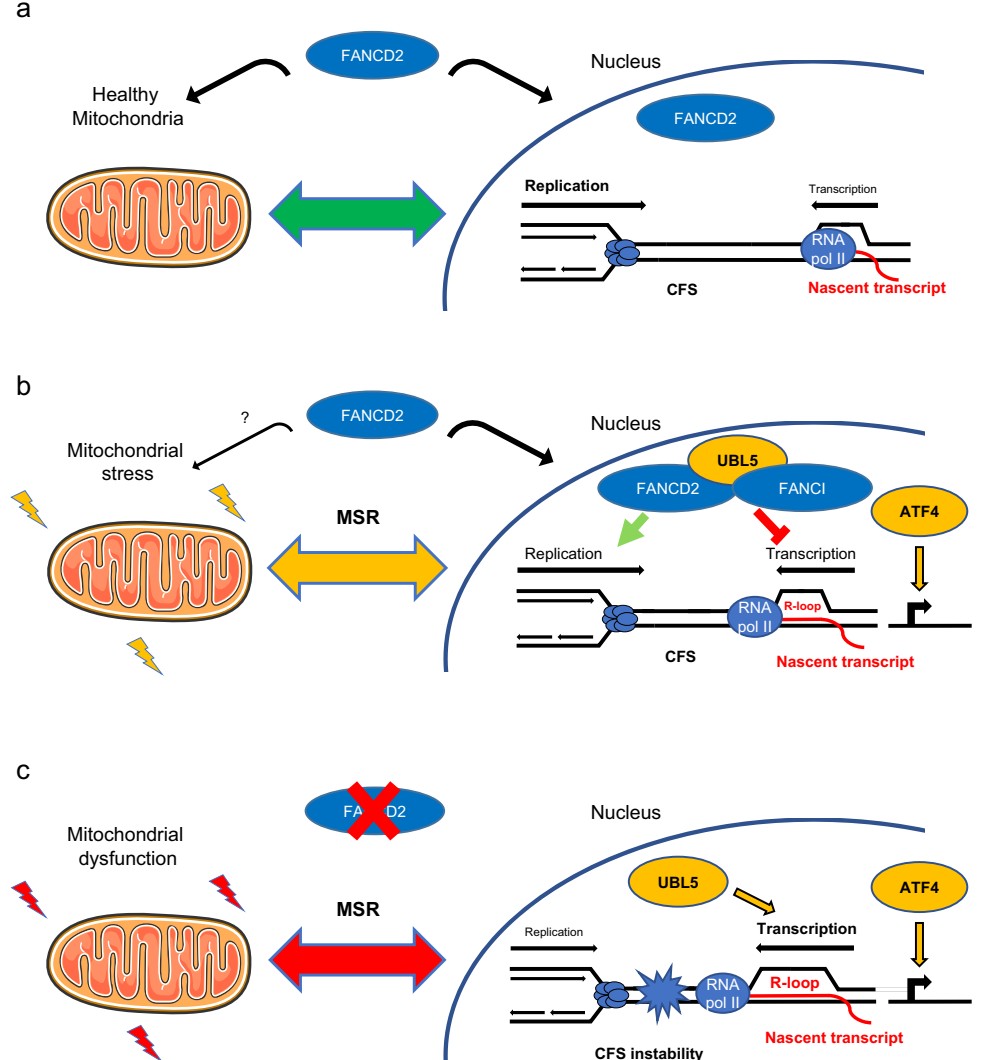

**Fig. 7 FANCD2 maintains mitochondrial homeostasis and genome stability by tuning the mitochondrial stress response. a** FANCD2 coordinates nuclear and mitochondrial activities to prevent mitochondrial dysfunction and maintain the mitochondrial stress response (MSR) in check. **b** Upon mitochondrial stress, FANCD2 relocalizes to CFS and dampens the UBL5-dependent mtUPR, limiting transcription–replication conflicts. **c** In the absence of FANCD2, mitochondrial dysfunction activates a mitochondrial stress response involving the ATF4 pathway and unrestrained UBL5-dependent CFS gene transcription, leading to transcription-replication collisions and CFS instability.

The ER or mitochondrial UPR was induced by treatment with 1 µM TG (Interchim 42759J) or 10 µM CCCP (Sigma C2759) for 8 h. For OXPHOS inhibition, sodium azide (NaN₃, Sigma S8032) was used at 20 mM for 10 h. For inhibition of the ISR, ISRIB (Sigma SML0843) was used at 500 nM for 8 h.

**Immunofluorescence**. Cells grown on glass cover slips were fixed in 4% for-maldehyde for 15 min before permeabilization with 0.5% Triton for 10 min at room temperature. After blocking with 3% BSA in PBS containing 0.05% Tween 20, cells were stained overnight with the primary antibody against FANCD2 and then with a secondary antibody, anti-rabbit Alexa Fluor 594 (Invitrogen), for 1 h at room temperature. Slides were mounted in DAKO mounting medium containing 4′,6-diamidino-2-phenylindole (DAPI) (SouthernBiotech) and examined at a ×63 magnification using an epifluorescence microscope (Zeiss Axio Observer Z1) equipped with an ORCA-ER camera (Hamamatsu). The microscope and camera parameters were set for each series of experiments to avoid signal saturation. Image processing and analysis were performed using ImageJ.

**Nascent transcript analysis**. Nascent transcripts were captured and analyzed using a Click-iT Nascent RNA Capture Kit from ThermoFisher following the manufacturer's instructions. Briefly, cells were seeded in a six-well plate and transfected the next day. Next, 48 h after transfection, cells were incubated with 0.5 mM EU for 1 h and harvested for RNA extraction. Then, 5 µg of RNA was biotinylated with 0.5 mM biotin azide and precipitated. Finally, 1 µg of biotinylated

RNA was bound to 50 µl of streptavidin magnetic beads and used for cDNA synthesis and qPCR.

**Metaphase spread preparation and FISH analysis**. Forty-eight hours after transfection, cells were incubated with or without 0.3 µM APH for 16 h. Subse-quently, the cells were exposed to 100 ng/ml colcemid (Roche) for 3 h, treated with hypotonic solution (0.075 M KCl) for 15 min and fixed with 3:1 ethanol/acetic acid overnight at −20 °C. The cells were then transferred onto slides and dried for one day. For FRA3B analysis, two FISH probes were used: the ZytoLight SPEC FHIT/CEN 3 dual color probe (ZytoLight) and labeled bacterial artificial chromosomes (BACs). Briefly, bacterial strains containing the BACs (RP11-170K19 and RP11-495E23) were grown overnight at 37 °C with 12.5 µg of chloramphenicol and extracted using a BACMAX DNA purification kit (Epicentre). Then, DNA was sonicated to obtain fragments shorter than 400 bp, which were then labeled green or red using a PlatinumBright labeling kit (Kreatech) following the manufacturer's instructions. A PARK2 FISH probe (Empire Genomics) was used for FRA6E analysis. Briefly, slides were sequentially incubated in 70%, 90% and 100% ethanol for 2 min and then dried. Subsequently, 10 µl of each probe was added to the slides, and a cover slip was added and adhered with rubber cement on its edges to avoid dehydration. The slides were placed on an automatic hybridizer (Hybridizer, Dako) and heated at 72 °C for 2 min and then at 37 °C for at least 16 h. Afterwards, the coverslips were removed in wash buffer (0.5× SSC and 0.1% SDS) at 37 °C, and the slides were incubated in wash buffer for 5 min at 65 °C to remove nonspecific

signals. Finally, the slides were washed with PBS, and DAPI was added with mounting medium for microscopic analysis.

**Chromatin immunoprecipitation and next-generation sequencing (ChIP-seq)**. ChIP-seq experiments were performed using Active Motif ChIP sequencing services. First, $1 \times 10^7$ HCT116 cells that had been treated with or without 0.3 μM APH were fixed in 1% formaldehyde for 15 min. After cell lysis, 30 μg of chromatin was used for immunoprecipitation using a FANCD2 antibody (Novus). Immunoprecipitated and input DNA were sequenced by Illumina sequencing, which generated 75-nt sequence reads. More than $30 \times 10^6$ reads per condition were obtained, and a spike-in adjusted normalization method was applied. The peaks were called using the SICER algorithm and aligned to the human genome build hg19. Integrative genomics viewer (IGV) was used to visualize peaks from the genome.

**ChIP and quantitative PCR**. After preclearing with magnetic beads for 1 h, the chromatin from an equivalent of $1 \times 10^7$ HCT116 cells was used for immunoprecipitation with a FANCD2 antibody (Novus) or immunoglobulin G as a control. After an overnight incubation at 4 °C, the beads were washed and eluted in buffer E (25 mM Tris–HCl [pH 7.5], 5 mM EDTA, and 0.5% SDS), and crosslinking was reversed at 65 °C with proteinase K for 6 h. The DNA was then purified using a QIAquick PCR purification kit (QIAGEN) and eluted in 100 μl of distilled water. The PCR primer pairs are listed in Supplementary Table S1.

**Oxygen consumption measurements**. Oxygen consumption was measured at 25 °C in a closed chamber using an amperometric electrode (Unisense Microrespiration, Unisense A/S, Denmark). Cells were permeabilized with 0.03 mg/ml digitonin for 1 min, centrifuged for 9 min at 1000×g and resuspended in a buffer containing 137 mM NaCl, 5 mM KCl, 0.7 mM KH₂PO₄, 25 mM Tris–HCl (pH 7.4), and 25 mg/ml ampicillin. The same solution was used in the oxymetric measurements. For each experiment, 500,000 cells were used. Finally, 10 mM pyruvate and 5 mM malate were added to stimulate the electron transfer pathway by complexes I, III and IV.

**Electron transfer from complex I to complex III**. Electron transfer from complex I to complex III was studied spectrophotometrically by following the reduction of cytochrome $c$ at 550 nm. The molar extinction coefficient used for reduced cytochrome $c$ was $1 \, \text{mM}^{-1} \, \text{cm}^{-1}$. For each assay, 50 μg of total protein was used. The assay medium contained 100 mM Tris–HCl (pH 7.4) and 0.03% cytochrome $c$. The reaction was initiated with the addition of 0.7 mM NADH. If electron transport between complexes I and III is conserved, the electrons pass from NADH to complex I, then to complex III via coenzyme Q, and finally to cytochrome $c$.

**ATP and AMP quantification**. ATP and AMP were measured according to the enzyme coupling method of Bergmeyer et al. (Bergmeyer HU, Grassl M, Walter HE (1983) Methods of Enzymatic Analysis, Verlag-Chemie, Weinheim, p. 249). For ATP assays, the medium contained 20 μg of sample, 50 mM Tris–HCl pH 8.0, 1 mM NADP, 10 mM MgCl₂, and 5 mM glucose in a final volume of 1 ml. Samples were analyzed spectrophotometrically before and after the addition of 4 μg of purified hexokinase/glucose-6-phosphate dehydrogenase (Boehringer). The decrease in absorbance at 340 nm due to NADPH formation was proportional to the ATP concentration. For AMP assays, the medium contained 20 μg of sample, 50 mM Tris–HCl (pH 8.0), 1 mM NADH, 10 mM MgCl₂, 10 mM phosphoenolpyruvate (PEP), and 2 mM ATP in a final volume of 1 ml. Samples were analyzed spectrophotometrically before and after the addition of 4 μg of purified pyruvate kinase/LDH (Boehringer). The decrease in absorbance at 340 nm due to NADH oxidation was proportional to the AMP concentration. For all biochemical experiments, protein concentrations were determined using the Bradford method.

**LDH activity assay**. LDH (EC 1.1.1.27) activity was measured to quantify the anaerobic metabolism. The reaction mixtures contained 100 mM Tris–HCl (pH 9), 5 mM pyruvate, 40 μM rotenone and 0.2 mM NADH, with LDH activity expressed as IU/mg of total protein (micromoles/min/mg of protein).

**Fo-F1 ATP synthase activity assay**. Evaluation of the Fo–F1 ATP synthase activity was performed as previously described. Briefly, 200,000 cells were incubated for 10 min in a medium containing 10 mM Tris–HCl (pH 7.4), 100 mM KCl, 5 mM KH₂PO₄, 1 mM EGTA, 2.5 mM EDTA, 5 mM MgCl₂, 0.6 mM ouabain and 25 mg/ml ampicillin, and 10 mM pyruvate plus 5 mM malate to stimulate the pathway composed by complexes I, III and IV. ATP synthesis was induced by the addition of 0.1 mM ADP. The reaction was monitored every 30 s for 2 min using a luminometer (GloMax® 20/20n Luminometer, Promega Italia, Milan, Italy) for the luciferin/luciferase chemiluminescent method, with ATP standard solutions used at concentrations between $10^{-8}$ and $10^{-5}$ M (luciferin/luciferase ATP bioluminescence assay kit CLSII, Roche, Basel, Switzerland). Data are expressed as nmol ATP produced/min/$10^6$ cells.

**Statistics and reproducibility**. All quantitative data are presented as the means ± SEM of at least three independent experiments. Unless otherwise stated, significance was tested using a two-tailed Student's $t$-test. Statistical tests were performed using Prism (GraphPad software). $p$ values are indicated as $*p \leq 0.05$, $**p < 0.01$, $***p < 0.001$, and $****p < 0.0001$, with ns indicating not significant ($p > 0.05$). Individual $p$ values are indicated in the figure legends.

**Reporting summary**. Further information on research design is available in the Nature Research Reporting Summary linked to this article.

## Data availability

Source data are available in Supplementary Data 1. All ChIP-seq data have been deposited in the National Center for Biotechnology Information Gene Expression Omnibus (GEO) and are accessible through the GEO Series accession number GSE141101. All other relevant data are available from the corresponding author on request.

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

## Acknowledgements

The authors are grateful to the members of UMR9019 CNRS for helpful discussions and advice, to M. Debatisse for the FHIT-KO cell line, and to M. Debatisse and F. Rosselli for critical reading of the manuscript. We thank N. Modjtahedi for the kind gift of the HSP60 antibody. The V. Naim laboratory was supported by a European Research Council Starting Grant (ERC-2014-StG-638898 "FAtoUnFRAGILITY") and a subvention from GEFLUC Paris Ile-de-France. The work of B.M. and C.S.-R. were supported by the Fondation pour la Recherche Médicale (AJE20151234749), the Institut National du Cancer-Plan Cancer (ASC15018KSA) and Labex "Who am I?" (ANR-11-LABX-0071 and ANR-11-IDEX-005-02). E.C. is indebted to AIRFA for its support in the activity of the Clinical & Experimental Hematology Unit of G. Gaslini Institute. P.F. was the recipient of a Ph.D. fellowship from University Paris-Sud. M.S. is the recipient of a Ph.D. fellowship from La Ligue contre le cancer.

## Author contributions

P.F. performed research, analyzed and interpreted data, and wrote the manuscript. B.M. and C.S.-R. performed the ChIP-seq analysis and ChIP-qPCR experiments. M.S. and V.B. contributed to data production and analysis. S.R. and E.C. performed biochemical and metabolic analyses. V. Nähse generated the FHIT-KO cell line. V. Naim conceived and supervised the project, carried out research, analyzed and interpreted data, and wrote the manuscript, with input from other authors.

## Competing interests

V. Naim is an Editorial Board Member for *Communications Biology*, but was not involved in the editorial review of, nor the decision to publish this article. The authors have no financial or other competing interests to declare.
