## [Peer Review File · Communications Biology]

Reviewers' comments:

Reviewer #1 (Remarks to the Author):

In this report, authors showed that activation of the mitochondrial unfolded protein response (UPR) pathway induces CFS expression and recruitment of FANCD2 to CFSs. FANCD2 suppresses CFS gene transcription and promotes CFS stability. FANCD2 also suppresses UPR activation and prevents mitochondrial dysfunction. UPR-dependent induction of CFS genes is mediated by UBL5. The finding that FANCD2 coordinates nuclear and mitochondrial activities through UPR to prevent genome instability is interesting, but some points need to be addressed to support the conclusions.

1. Does mitochondrial dysfunction and UPR induce general nuclear DNA damage responses such as ATR activation, RPA phosphorylation and DSB formation?
2. Does mitochondrial dysfunction and UPR induce R-loop formation genome-wide? CFSs may be a major hit. Does R-loop also form in mitochondria especially at the genes FANCD2 finds?
3. Reconstitution of FANCD2 in FANCD2-deficient cells is needed to show the specificity.
4. Does FANCD2 associated protein FANCI play a similar role in mediating UPR-induced CFS gene expression and protection as well as suppression of UPR and mitochondrial dysfunction. Does FANCI bind to CFSs in a manner similar to FANCD2?
5. If mapped MURE1, CHOP and MURE2 elements at CFSs are deleted, will the stability of CFSs and gene expression at CFSs increase?
6. Does FANCD2 knockdown induce UBL5 overexpression in mammalian cells? Upon UPR, what is the mechanism of how UBL5 cause CFS gene expression and instability?

Reviewer #2 (Remarks to the Author):

The manuscript by Fernandes et al. describes a functional link between the FANCD2-regulated expression of common fragile site (CFS) – harboring genes and the mitochondrial unfolded protein response, UPR_{mt}. The authors show that FANCD2 prevents breakage at CFS-containing genes in a transcription-dependent manner. The authors identify UPR_{mt} elements within CFS genes as preferential FANCD2 binding sites, and demonstrate that UPR_{mt} induction triggers CFS gene expression, FANCD2 recruitment and response modulation. The authors then demonstrate that FANCD2 regulates CFS gene expression by counteracting the UPR_{mt} factor, UBL5. Lastly, the authors further demonstrate a new role for FANCD2 in promoting mitochondrial oxidative phosphorylation (OXPHOS) and functionally link OXPHOS to CFS breakage and gene expression.

Overall, the paper is well written per se and has some interesting new findings that functionally link FANCD2 to mitochondrial function and maintenance in the context of CFS gene expression. However, I have some conceptual difficulties with the working model, based on the data presented in the manuscript. I feel that these need to be addressed before I can recommend publication of the manuscript in Communications Biology.

Main concerns:

- 1) The authors show that FANCD2-deficient cells contain significantly increased mRNA and protein levels of FHIT and other CFS genes. Simultaneously, FANCD2-depleted cells exhibit mitochondrial defects such as reduced OXPHOS. How do the authors reconcile these data with their other finding that OXPHOS is required for CFS gene expression? The two findings seem to directly contradict one another.

- 2) Similar to concern 1): The authors use CCCP to induce an UPRmt. CCCP is a known mitochondrial uncoupler that leads to reduced ATP synthesis. The authors show that cellular CCCP treatment causes an increase in CFS gene expression, whereas use of another OXPHOS inhibitor (i.e. blocker of ATP synthesis), sodium azide, has the opposite effect on CFS gene expression. Why?
- 3) In my opinion, the logical flow of the manuscript is somewhat interrupted by the sudden switch to oxidative phosphorylation in between, without further exploration of how (and why) the UPRmt signaling pathway induces FHIT/other CSF gene expression in a FANCD2-regulated manner.
- 4) Based on the finding that CSF gene expression in FANCD2-deficient cells can be further upregulated by treatment with CCCP or TG, the authors propose that FANCD2 'dampens' the CFS gene response to UPRmt signaling. It would be nice to see this idea supported by an experiment where FANCD2 overexpression blocks the UPRmt-inducible CFS gene expression.
- 5) Unlike CCCP and TG, cellular treatment with siSPG7 triggers CFS gene upregulation independently of FANCD2. Possible reasons for this should be discussed.

Minor comments:

- Fig 1: it would be easier on the eye to have the actual relative increases in FHIT protein expression levels provided as numbers right below the corresponding western blot bands in Figs. 1D, 1F and 1H.
- Fig 2 A has no error bars
- Fig 2C: are these differences significant?
-

Reviewer #3 (Remarks to the Author):

In their manuscript, the authors describe a connection between common fragile site (CFS) instability and protein stress signaling in cells. Using FANCD2 knockdown, they showed increased transcription of CFS genes, as well as increased CFS instability, due to a loss of FANCD2 targeting to CFSs. Analysis of FANCD2 binding regions by ChIP-seq revealed localization of mitochondrial unfolded protein response (mtUPR) elements in some of these, suggesting a potential connection between FANCD2 and mtUPR. Upon induction of mitochondrial and ER stresses, FANCD2 localizes to CFS genes, which become induced. Finally, the authors show that mitochondrial respiration defects can be induced by FANCD2 deficiency and are capable of inducing CFS genes. These effects are affected by ATF4 knockdown, but not depletion. Overall, the manuscript is interesting and well presented. Indeed, gaining and understanding of possible connections between these two cellular pathways would be important. While the shown results on the connection of FANCD2 with CFS looks convincing to me, there are still a number of open questions regarding the link to stress responses, largely driven by unclear descriptions.

Major points:

- 1) The used nomenclature regarding stress responses is generally unprecise and distracts from the understanding of the manuscript. In its current state, it's incorrect in many places. Unfolded protein response is a defined term and refers to the ER stress response only. Mitochondrial UPR (mtUPR or UPRmt) only describes responses to mitochondrial protein misfolding. Generally, unfolded protein responses refer to stresses that are caused by proteostasis defects (i.e. misfolding/unfolding). It is often not clear to me what the authors are referring to.
- 2) CCCP is not a UPR inducer, it induces mitochondrial stress (breakdown of the mitochondrial

membrane potential). The pathway linking ER stress and mitochondrial stress, and potentially explaining the observed results shared by CCCP and thapsigargin treatments, is the integrated stress response (ISR), which is activated by both treatments. This pathway is likely at the center of the effects observed in this manuscript. In that regard, the author need to test whether they see mRNA induction of specific targets by the different responses, i.e. HSP60/10 for mtUPR, ATF4 and CHOP for ISR, BIP for ER stress. That would clarify what response pathway is causing the observed effects. In the likely case of it being mediated by the ISR, controls using ISRIB are required that would allow to determine whether a block of ISR effects is sufficient to see changes in CFS gene induction.

3) In Mammalian cells, the role of UBL5 in the mtUPR has not been shown. While its knockdown/KO clearly has an effect on CFS genes similar to FANCD2, the interpretation in regards to the mtUPR cannot be drawn. Here, it would be required to establish what signaling is referred to (ISR versus mtUPR) and to determine the effects of the respective targets of these pathways (ATF4/CHOP versus HSP60/10, respectively). How does UBL5 knockdown affect these pathways and does UBL5 affect these stress pathways in the context of the described stress conditions (i.e. CCCP, thapsigargin).

Minor points

1) The statement "The mammalian mitochondrial UPR can be triggered by stress in the mitochondria or ER42" is unclear and I would remove it. The mitochondrial stress response is triggered by mitochondrial stress. Large parts of the literature is derived from data in *C. elegans* that has proven to not be conserved in mammalian cells. The same applies to the use of reference 50 and the use of SPG7 knockdowns in mammalian cells.

Referee expertise:

Referee #1:

Referee #2:

Referee #3:

Reviewers' comments:

Reviewer #1 (Remarks to the Author):

In this report, authors showed that activation of the mitochondrial unfolded protein response (UPR) pathway induces CFS expression and recruitment of FANCD2 to CFSs. FANCD2 suppresses CFS gene transcription and promotes CFS stability. FANCD2 also suppresses UPR activation and prevents mitochondrial dysfunction. UPR-dependent induction of CFS genes is mediated by UBL5. The finding that FANCD2 coordinates nuclear and mitochondrial activities through UPR to prevent genome instability is interesting, but some points need to be addressed to support the conclusions.

We thank the reviewer #1 for appreciation of the interest of our finding.

1. Does mitochondrial dysfunction and UPR induce general nuclear DNA damage responses such as ATR activation, RPA phosphorylation and DSB formation?

We thank the reviewer for raising this question, we have now addressed this point and show that UPR activation does not induce a general nuclear DNA damage response, as assessed by phosphorylation of ATR targets chk1 and RPA, and gammaH2AX foci formation. We have now included these results in the revised version.

2. Does mitochondrial dysfunction and UPR induce R-loop formation genome-wide? CFSs may be a major hit. Does R-loop also form in mitochondria especially at the genes FANCD2 finds?

Testing R-loop formation genome-wide with approaches like DRIP-seq for nuclear or mitochondrial DNA would certainly be interesting, but we think this is beyond the aim of the present manuscript. We agree with the reviewer that CFSs are major hits, as indicated by our ChIP-seq analysis of FANCD2 binding sites showing that FANCD2 targets CFSs as well as other sites enriched in MURE elements.

3. Reconstitution of FANCD2 in FANCD2-deficient cells is needed to show the specificity.

We have tried the reconstitution with a plasmid over-expressing an siRNA-resistant FANCD2, but due to experimental difficulties we did not succeed. To exclude any off-target effects, we have used several different FANCD2-specific siRNA that have been validated in previous publications (see attached Annex: different siRNAs). Please note that we have also tested the effects of depletion of FANCA and FANCI, which gives similar results, even though upregulation of CFS genes, at least at the mRNA level, is lesser.

4. Does FANCD2 associated protein FANCI play a similar role in mediating UPR-induced CFS gene expression and protection as well as suppression of UPR and mitochondrial dysfunction. Does FANCI bind to CFSs in a manner similar to FANCD2?

We have analyzed CFS gene expression and show that, similar to what we observed with FANCD2, FANCI also attenuates CFS gene expression. In addition, we have verified by CHIP-qPCR its recruitment to CFSs after replication stress and after UPR induction by CCCP or TG. We have now included part of these data in the revised version of the manuscript. We planned to test the role of FANCI on mitochondrial function but, due to difficulties linked to the lockdown, we could not achieve these data. This would also help to determine the respective roles of FANCD2 and FANCI in the mitochondria and/or in the nucleus, but we think it could be the subject of a follow-up study.

5. If mapped MURE1, CHOP and MURE2 elements at CFSs are deleted, will the stability of CFSs and gene expression at CFSs increase?

We agree that this would be very interesting to test, but since there are many MURE and CHOP elements in CFS genes, it would require targeting several different elements alone or in combination. Sequencing and analysis of clones harboring deletions and copy number variations in CFS genes should help in addressing this issue and examine how the ability of cells to respond to the mtUPR is affected, as we proposed in the discussion, but this might be the subject of future study.

6. Does FANCD2 knockdown induce UBL5 overexpression in mammalian cells? Upon UPR, what is the mechanism of how UBL5 cause CFS gene expression and instability?

We thank the referee for this suggestion. Indeed, we have found a slight but reproducible induction of UBL5 after FANCD2 knockdown (see attached Annex: UBL5). The mechanism of induction of CFS gene expression and instability is (co-)transcriptional, because depleting UBL5 or inhibiting transcription reduces breakage. Please note that UBL5 is a conserved non-covalent component and modulator of the spliceosome (Oka Y., et al, EMBO Rep, 2014) and regulates FANCI pre-mRNA splicing and translation in addition to protein stability and interaction with FANCD2 (Oka Y., et al., EMBO J, 2015).

Reviewer #2 (Remarks to the Author):

The manuscript by Fernandes et al. describes a functional link between the FANCD2-regulated expression of common fragile site (CFS) – harboring genes and the mitochondrial unfolded protein response, UPR_{mt}. The authors show that FANCD2 prevents breakage at CFS-containing genes in a transcription-dependent manner. The authors identify UPR_{mt} elements within CFS genes as preferential FANCD2 binding sites, and demonstrate that UPR_{mt} induction triggers CFS gene expression, FANCD2 recruitment and response modulation. The authors then demonstrate that FANCD2 regulates CFS gene expression by counteracting the UPR_{mt} factor, UBL5. Lastly, the authors further demonstrate a new role for FANCD2 in promoting mitochondrial oxidative phosphorylation (OXPHOS) and functionally link OXPHOS to CFS breakage and gene expression.

Overall, the paper is well written per se and has some interesting new findings that functionally link FANCD2 to mitochondrial function and maintenance in the context of CFS gene expression.

We thank the reviewer for her/his positive comments on our paper.

However, I have some conceptual difficulties with the working model, based on the data presented in the manuscript. I feel that these need to be addressed before I can recommend publication of the manuscript in Communications Biology.

Main concerns:

1) The authors show that FANCD2-deficient cells contain significantly increased mRNA and protein levels of FHIT and other CFS genes. Simultaneously, FANCD2-depleted cells exhibit mitochondrial defects such as reduced OXPHOS. How do the authors reconcile these data with their other finding that OXPHOS is required for CFS gene expression? The two findings seem to directly contradict one another.

Indeed, FANCD2 deficiency causes a decrement of mitochondrial ATP synthesis, but an increase in oxygen consumption, as reported in Figure 4 (previous Figure 5), Panel A. This determines an impairment of energy production, but also enhances the oxidative stress by the increment of the uncontrolled respiration rate. Therefore, the partial function of OXPHOS could be sufficient to activate the CFS gene expression.

2) Similar to concern 1): The authors use CCCP to induce an UPRmt. CCCP is a known mitochondrial uncoupler that leads to reduced ATP synthesis. The authors show that cellular CCCP treatment causes an increase in CFS gene expression, whereas use of another OXPHOS inhibitor (i.e. blocker of ATP synthesis), sodium azide, has the opposite effect on CFS gene expression. Why?

This apparent discrepancy can be explained considering the different effects of CCCP and sodium azide on OXPHOS activity.

OXPHOS machinery is composed of two parts: the electron transport chain (ETC), devoted to oxygen consumption, and the Fo-F1 ATP synthase, which produces ATP by the proton gradient formed by ETC. In physiological conditions, these two parts work synergically (coupled status), and the block of ATP synthesis determines the slowdown of ETC. By contrast, in the uncoupled status, the activity of ATP synthesis is not associated with respiration and the inhibition of ATP synthase does not affect the oxygen consumption rate.

Therefore, we have observed the increase of CFS gene expression with CCCP, but not with sodium azide because the first is an uncoupling molecule, while the second is a direct inhibitor of ATP synthase.

In particular, CCCP, preventing the proton gradient formation across the mitochondrial inner membrane, impairs the ATP synthesis, causing an acceleration of the respiration rate. This determines an increment of oxidative stress production and the consequent damage to protein folding. Conversely, sodium azide, inhibiting the ATP synthesis without an uncoupling effect, determines also a reduction of oxygen consumption, diminishing the risk of oxidative stress production and relative damages.

Therefore, since our data indicate that the increase of CFS gene expression depends on the accumulation of the unfolded proteins, which could be associated with an uncontrolled oxidative

stress production, it is possible to suggest that the mitochondrial activity inhibitors mimic the FANCD2 deficiency, only when acting as uncoupling molecules.

3) In my opinion, the logical flow of the manuscript is somewhat interrupted by the sudden switch to oxidative phosphorylation in between, without further exploration of how (and why) the UPRmt signaling pathway induces FHIT/other CSF gene expression in a FANCD2-regulated manner.

We apologize for this logical flow interruption. We have now partially reorganized the text and moved the analysis of OXPHOS before. We hope that now the logical flow is improved.

4) Based on the finding that CSF gene expression in FANCD2-deficient cells can be further upregulated by treatment with CCCP or TG, the authors propose that FANCD2 'dampens' the CFS gene response to UPRmt signaling. It would be nice to see this idea supported by an experiment where FANCD2 overexpression blocks the UPRmt-inducible CFS gene expression.

We have overexpressed FANCD2 and found that it does not block the UPRmt-inducible CFS expression (see attached Annex: FANCD2 overexpression). This is not surprising because the expression, stability, DNA binding and assembly of the FANCD2-FANCI (ID) complex on the chromatin is tightly regulated. For instance, D2-I heterodimerization is regulated by UBL5 (Oka Y et al., EMBO J, 2015); the chromatin binding and ID complex dosage upon DNA damage and during the cell cycle is regulated by mono-ubiquitylation and deubiquitylation (Nijman SM, et al., Mol Cell. 2005), SUMOylation and ubiquitylation (Gibbs-Seymour I et al., Mol Cell, 2015). In addition, FANCD2 is dynamically recruited at sites of encounters between transcription and replication machineries and FANCD2 accumulation is largely R-loop dependent (Schwab RA et al., Mol Cell, 2015; García-Rubio ML et al., PLoS, 2015).

5) Unlike CCCP and TG, cellular treatment with siSPG7 triggers CFS gene upregulation independently of FANCD2. Possible reasons for this should be discussed.

This can be explained by the fact that SPG7 regulates protein turnover and respiratory chain complex assembly in the inner mitochondrial membrane, thus its depletion leads to the accumulation of unfolded or unassembled proteins independently of FANCD2 function, which is involved in upstream (mitochondrial protein biogenesis) or downstream (mitophagy regulation) steps of mitochondrial biogenesis and quality control; supporting this view, we have further determined that FANCD2 knockdown induces *SPG7* transcription (similar to what reported for depletion of *COQ7*, another gene suppressing the mtUPR (Monaghan RM et al., Nat Cell Biol, 2015), which may be necessary to get rid of incorrectly folded or damaged proteins produced in absence of FANCD2.

Minor comments:

- Fig 1: it would be easier on the eye to have the actual relative increases in FHIT protein expression levels provided as numbers right below the corresponding western blot bands in Figs. 1D, 1F and 1H.

We have now added the quantifications in Figure 1.

- Fig 2 A has no error bars

Here we just wanted to show the complete absence of *FHIT* expression in the FHIT-KO cells.

- Fig 2C: are these differences significant?

We have now provided the statistical information showing a significant decrease in FANCD2 binding to *FHIT* in FHIT-KO cells.

Reviewer #3 (Remarks to the Author):

In their manuscript, the authors describe a connection between common fragile site (CFS) instability and protein stress signaling in cells. Using FANCD2 knockdown, they showed increased transcription of CFS genes, as well as increased CFS instability, due to a loss of FANCD2 targeting to CFSs. Analysis of FANCD2 binding regions by ChIP-seq revealed localization of mitochondrial unfolded protein response (mtUPR) elements in some of these, suggesting a potential connection between FANCD2 and mtUPR. Upon induction of mitochondrial and ER stresses, FANCD2 localizes to CFS genes, which become induced. Finally, the authors show that mitochondrial respiration defects can be induced by FANCD2 deficiency and are capable of inducing CFS genes. These effects are affected by ATF4 knockdown, but not depletion. Overall, the manuscript is interesting and well presented. Indeed, gaining and understanding of possible connections between these two cellular pathways would be important. While the shown results on the connection of FANCD2 with CFS looks convincing to me, there are still a number of open questions regarding the link to stress responses, largely driven by unclear descriptions.

We thank the reviewer for her/his positive comments on our manuscript. We have now tried to clarify the relative descriptions and performed new experiments to support the implication of FANCD2 and UBL5 in the mtUPR. We think that these new data show that FANCD2 suppress the mtUPR and respond to it when it is activated. The fact that some of the responses can be induced by both mitochondrial and ER stress may be due to crosstalk between organelles and the fact that the mitochondrial stress response is embedded in the ISR that can be activated by both CCCP and TG treatments (Anderson NS and Haynes CM, Trends in Cell Biol., 2020).

Major points:

1) The used nomenclature regarding stress responses is generally unprecise and distracts from the understanding of the manuscript. In its current state, it's incorrect in many places. Unfolded protein response is a defined term and refers to the ER stress response only. Mitochondrial UPR (mtUPR or UPRmt) only describes responses to mitochondrial protein misfolding. Generally, unfolded protein responses refer to stresses that are caused by proteostasis defects (i.e. misfolding/unfolding). It is often not clear to me what the authors are referring to.

We thank the reviewer for this criticism. We have now taken this into account and referred to the mtUPR.

2) CCCP is not a UPR inducer, it induces mitochondrial stress (breakdown of the mitochondrial membrane potential). The pathway linking ER stress and mitochondrial stress, and potentially

explaining the observed results shared by CCCP and thapsigargin treatments, is the integrated stress response (ISR), which is activated by both treatments. This pathway is likely at the center of the effects observed in this manuscript. In that regard, the author need to test whether they see mRNA induction of specific targets by the different responses, i.e. HSP60/10 for mtUPR, ATF4 and CHOP for ISR, BIP for ER stress. That would clarify what response pathway is causing the observed effects. In the likely case of it being mediated by the ISR, controls using ISRIB are required that would allow to determine whether a block of ISR effects is sufficient to see changes in CFS gene induction.

We thank again the referee for this important point. We have now performed additional experiments and show that indeed, even if CCCP and TG induce the ISR, CFS induction is clearly distinct from the ISR and is regulated by the UBL5-dependent mtUPR.

We show that CHOP induction is dependent on both the ISR and FANCD2, and that CFS genes respond to UBL5 mediated mtUPR and not ISR. We think that these data unequivocally demonstrate the functional interaction of FANCD2 and UBL5 in the mtUPR in mammalian cells. We have not found significant induction of HSP60/10 mRNA. This may be due to a cell-type or tissue specific regulation HSP60/10 (Bahat A, et al., *Molecular Endocrinology* 28: 208–224, 2014). Notably HSP60/10 expression is increased at the protein level in *Fancd2* KO mice (Chatla S et al., *Stem Cell Res*, 2019).

3) In Mammalian cells, the role of UBL5 in the mtUPR has not been shown. While its knockdown/KO clearly has an effect on CFS genes similar to FANCD2, the interpretation in regards to the mtUPR cannot be drawn. Here, it would be required to establish what signaling is referred to (ISR versus mtUPR) and to determine the effects of the respective targets of these pathways (ATF4/CHOP versus HSP60/10, respectively). How does UBL5 knockdown affect these pathways and does UBL5 affect these stress pathways in the context of the described stress conditions (i.e. CCCP, thapsigargin).

We agree with the reviewer that the implication of UBL5 in the mtUPR in mammalian cells is shown here for the first time. We show that FANCD2 and UBL5 significantly affect the expression of CFS genes and CHOP. Noticeably, CHOP is the first mtUPR gene identified in mammalian cells (Zhao Q. et al., *EMBO J*, 2002).

Minor points

1) The statement “The mammalian mitochondrial UPR can be triggered by stress in the mitochondria or ER42” is unclear and I would remove it. The mitochondrial stress response is triggered by mitochondrial stress. Large parts of the literature is derived from data in *C. elegans* that has proven to not be conserved in mammalian cells. The same applies to the use of reference 50 and the use of SPG7 knockdowns in mammalian cells.

We have modified this statement and hope it is clearer like this. As for the SPG7 knockdown, we think that our data show that it is also involved in mtUPR in mammalian cells. Indeed, a previous work has shown that its expression is induced, together with other mtUPR genes, in response to the expression of a mutant COQ7, defective in nuclear targeting (Monaghan RM et al., *Nat Cell Biol*, 2015).

Fernandes et al.

Annex: Tested siRNA sequences targeting FANCD2

siRNA sequences

siRNA	Sequence	Reference
FANCD2	GGA-GAU-UGA-UGG-UCU-ACU-A	Bourseguin J et al. Scientific reports . 2016;6: 36539
FANCD2 #2	GAU-AAG-UUG-UCG-UCU-AUU-A	Truong LN et al. JBC . 2014;289(42): 28910-23
FANCD2 #18	GAA-CAA-AGG-AAG-CCG-GAA-U	Chen X. et al. Oncogene . 2016;35(1): 22-34.
FANCD2 #CELL	CAG-AGU-UUG-CUU-CAC-UCU-CUA	Kratz K. et al. Cell . 2010;142(1):77-88

FHIT mRNA levels after transfection with different siFANCD2 RNAs.

Fernandes et al.
Annex: UBL5

Fernandes et al.
Annex: FANCD2 overexpression

Reviewers' comments:

Reviewer #1 (Remarks to the Author):

The authors addressed some reviewers' comments with new experimental data but left some other points for future studies. While I agree some points can be addressed in the future, two specific and straightforward experiments need to be completed for this study. First, FANCD2 reconstitution (comment 3) is a pretty routine experiment, which has been done in a number of publications. Second, testing the role of FANCI on mitochondrial function needs to be completed (comment 4). I fully understand the difficulties due to the COVID-19 lockdown, and I recommend the editor to allow more time to complete the experiments.

Reviewer #2 (Remarks to the Author):

The authors have made every effort to answer my remaining concerns and questions, and the manuscript is in my opinion now appropriate for publication in Communications Biology.

I would make one final suggestion regarding Fig 1 d, f, h and i: Please place the protein level quantification numbers immediately underneath the FHIT blots in the respective figures, and make sure the numbers are written as 8.9, not as 8,9, etc.

Reviewer #3 (Remarks to the Author):

The manuscript is strongly improved and experimentally, all questions have been addressed sufficiently. However, I strongly disagree with the interpretation of the data in regards to whether this constitutes a mtUPR.

Opposite to how the authors state Zhao et al. EMBO 2002, the abstract of Zhao et al mentions "We find that the accumulation of unfolded protein within the mitochondrial matrix results in the transcriptional upregulation of nuclear genes encoding mitochondrial stress proteins such as chaperonin 60, chaperonin 10, mtDnaJ and ClpP, but not those encoding stress proteins of the endoplasmic reticulum.". Throughout the paper, Zhao et al. monitor chaperonin (i.e. HSP60/HSPD1 and HSP10/HSPA1) induction as readout of the mtUPR. CHOP has an important role as one of the transcription factors involved, but it is not a readout for mtUPR. CHOP is induced by ISR activation and thus considered a readout for the ISR. (As a side-note the Martinus et al 1996 FEBS from Hoogenraad is generally considered the first mtUPR paper and also monitors chaperonin activation). In summary, chaperonin induction is an essential marker for mtUPR induction in mammalian cells, CHOP or ATF4 induction depict ISR induction.

In my opinion, the authors observe mitochondria-induced induction of the ISR. This may be called mitochondrial stress response (MSR), ISR or ISRmt (as in Khan et al Cell Metab 2017, PMID: 28768179), but not mtUPR. The use of mtUPR is wrong and needs to be changed throughout the manuscript including the title. The correct description as ISR instead of mtUPR does not take any novelty away from the manuscript and it remains of sufficient interest for publication. Thus, once the authors make it sufficiently clear in the manuscript that the signalling goes to the ISR and this is affected by FANCD2, I fully support publication of the manuscript.

Please find below (in blue) a point by point response to the reviewers' comments.

Reviewers' comments:

Reviewer #1 (Remarks to the Author):

The authors addressed some reviewers' comments with new experimental data but left some other points for future studies. While I agree some points can be addressed in the future, two specific and straightforward experiments need to be completed for this study. First, FANCD2 reconstitution (comment 3) is a pretty routine experiment, which has been done in a number of publications. Second, testing the role of FANCI on mitochondrial function needs to be completed (comment 4). I fully understand the difficulties due to the COVID-19 lockdown, and I recommend the editor to allow more time to complete the experiments.

We understand the Reviewer #1 comment, we apologize not to be able to complete all the requested experiments.

Reviewer #2 (Remarks to the Author):

The authors have made every effort to answer my remaining concerns and questions, and the manuscript is in my opinion now appropriate for publication in Communications Biology.

I would make one final suggestion regarding Fig 1 d, f, h and i: Please place the protein level quantification numbers immediately underneath the FHIT blots in the respective figures, and make sure the numbers are written as 8.9, not as 8,9, etc.

We thank the reviewer #2 for her/his positive comments and suggestion. We have moved the quantifications and corrected the numbers as indicated.

Reviewer #3 (Remarks to the Author):

The manuscript is strongly improved and experimentally, all questions have been addressed sufficiently. However, I strongly disagree with the interpretation of the data in regards to whether this constitutes a mtUPR.

Opposite to how the authors state Zhao et al. EMBO 2002, the abstract of Zhao et al mentions "We find that the accumulation of unfolded protein within the mitochondrial matrix results in the transcriptional upregulation of nuclear genes encoding mitochondrial stress proteins such as chaperonin 60, chaperonin 10, mtDnaJ and ClpP, but not those encoding stress proteins of the endoplasmic reticulum." Throughout the paper, Zhao et al. monitor chaperonin (i.e. HSP60/HSPD1 and HSP10/HSPE1) induction as readout of the mtUPR. CHOP has an important role as one of the transcription factors involved, but it is not a readout for mtUPR. CHOP is induced by ISR activation and thus considered a readout for the ISR. (As a side-note the Martinus et al 1996 FEBS from Hoogenraad is generally considered the first mtUPR paper and also monitors chaperonin activation). In summary, chaperonin induction is an essential marker for mtUPR induction in mammalian cells, CHOP or ATF4 induction depict ISR induction.

In my opinion, the authors observe mitochondria-induced induction of the ISR. This may be called mitochondrial stress response (MSR), ISR or ISRmt (as in Khan et al Cell Metab 2017, PMID:

28768179), but not mtUPR. The use of mtUPR is wrong and needs to be changed throughout the manuscript including the title. The correct description as ISR instead of mtUPR does not take any novelty away from the manuscript and it remains of sufficient interest for publication. Thus, once the authors make it sufficiently clear in the manuscript that the signalling goes to the ISR and this is affected by FANCD2, I fully support publication of the manuscript.

We thank the reviewer #3 for the positive comments on our work, and for drawing our attention on the paper of Khan et al Cell Metab 2017. Here a defect in Twinkle activates the ISRmt that comprises both the ATF4 pathway and the mtUPR, which is consistent with what we observe after depletion of FANCD2. We have then taken into account the point raised by the reviewer on the fact that chaperonin expression is an essential marker for mtUPR induction. In fact, we could not find induction of these genes at the mRNA level after depletion of FANCD2 in our cells. Since however induction of Hsp60, Hsp10, as well as mtDnaJ and ClpP has been shown in hematopoietic cells of Fancd2 KO mice, we reasoned that this could be due to cell-type/tissue specific differences and/or to the fact that the induction is dependent on the level of unfolded proteins that accumulate in the mitochondria and the duration of the response, as reported in Zhao et al., EMBO J, 2002 and in Seiferling D. et al., EMBO Rep., 2016. In the latter work for example there is a difference in the expression of UPRmt markers (Hsp60 and mtDnaJ) in the hearts of DARS2 KO mice at 3-4 weeks compared to 6-weeks old mice; ClpP appears down-regulated, contrary to what observed in mice with Twinkle mutations; and ClpP deficiency, which induces on its own mitochondrial stress, induces a different response in mouse hearts *in vivo* with respect to HEK293T cells, notably HSP60 is slightly induced and Dnaja3 is not in 6-weeks mice, while DNAJA3 but not HSP60 is significantly induced in HEK293T cells; in addition in ClpP KO mice, mtHSP70 expression is induced at the protein level but not at the mRNA level. This shows that different subsets of mt stress response genes may be induced in a cell type or time-dependent manner and differentially regulated at the protein and transcript level.

Notably, we have tested HSP60 protein levels and found a slight increase in FANCD2 depleted cells. In addition, FANCD2 binds at the bidirectional promoter of HSPD1 and HSPE1 (these data are now shown as Supplementary Fig. 5b, c, d in the revised version of the manuscript). This suggests a role of FANCD2 in their regulation, which may be transcript level- and R-loop- dependent, as for CFS genes. Therefore, we believe that our data are consistent with a role of FANCD2 (and FANCI) in tuning the mtUPR, which allows to synchronize nuclear and mitochondrial activities. The absence of FANCD2 induces a mitochondrial stress response, similar to the ISRmt, involving the ATF4 pathway and a UBL5-dependent mtUPR, which cannot be correctly modulated and coordinated with replication, generating transcription-associated replication stress and genome instability.

We have modified the text to integrate the reviewer remarks and clarify these points better (all modifications are tagged in blue), we hope this will help to avoid any incorrect interpretation.

Reviewers' comments:

Reviewer #3 (Remarks to the Author):

Experimentally, I think it is a good manuscript. It remains hard to follow why the authors do not change their wording (and some interpretation) to what is standard in the proteostasis field. For me as a proteostasis researcher, wrong semantics and concepts make this manuscript very hard to read and to follow the conclusions as stated:

i) In the abstract and general text, the authors still write UPR to refer to mitochondrial unfolded protein response. UPR is a very well established abbreviation that always means ER stress (never mitochondrial stress). It cannot be used for mitochondrial UPR without adding a nominator such as mt (i.e. mtUPR or UPRmt) as everyone in the field will otherwise be confused (as am I).

ii) The authors write "We have then taken into account the point raised by the reviewer on the fact that chaperonin expression is an essential marker for mtUPR induction. In fact, we could not find induction of these genes at the mRNA level after depletion of FANCD2 in our cells." Thus, the authors do not see induction of the UPRmt! If the markers of the stress are not induced, by general definition in the field, the stress is not induced. It appears irrelevant in this context what effects are described in other papers for studies of mice with mutations in different genes that directly lead to mitochondrial misfolding. I would advise the authors to refer to ISR(mt) or mitochondrial stress response rather than UPRmt as UPRmt markers are not shown to change.

iii) Indeed, the authors write "The absence of FANCD2 induces a mitochondrial stress response, similar to the ISRmt, involving the ATF4 pathway and a UBL5-dependent mtUPR, which cannot be correctly modulated and coordinated with replication, generating transcription-associated replication stress and genome instability." This statement nearly exactly entails what I would like to convey to the authors in that what they observe is a "mitochondrial stress response, similar to the ISRmt". This should be the main message of the paper/abstract/model. Despite of the lack of data on a UPRmt induction, the link to UBL5 and possibly UPRmt is reasonable to be made, but cannot be the focus of the paper due to lack of supporting data. If the authors were to change title/manuscript/model accordingly, I would strongly suggest publication.

Reviewer #4

I've read over the manuscript by Fernandes et al. and I tend to agree with Reviewer #3:

i) In the abstract and general text, the authors still write UPR to refer to mitochondrial unfolded protein response. UPR is a very well established abbreviation that always means ER stress (never mitochondrial stress). It cannot be used for mitochondrial UPR without adding a nominator such as mt (i.e. mtUPR or UPRmt) as everyone in the field will otherwise be confused (as am I).

<< I too found the use of "UPR" confusing. The precise use of "UPR" historically refers to the stress response pathway originating from ER stress. The UPR resulting from mitochondrial stress must be referred to as mitochondrial UPR or an abbreviated form (e.g. UPRmt).

ii) The authors write "We have then taken into account the point raised by the reviewer on the fact that chaperonin expression is an essential marker for mtUPR induction. In fact, we could not find induction of these genes at the mRNA level after depletion of FANCD2 in our cells." Thus, the authors do not see induction of the UPRmt! If the markers of the stress are not induced, by general definition

in the field, the stress is not induced. It appears irrelevant in this context what effects are described in other papers for studies of mice with mutations in different genes that directly lead to mitochondrial misfolding. I would advise the authors to refer to ISR(mt) or mitochondrial stress response rather than UPRmt as UPRmt markers are not shown to change.

<< Indeed, the UPRmt is classically associated with induction of these genes at the mRNA level. I agree with the reviewer to refer their findings as an ISRmt or mitochondrial stress response rather than the UPRmt.

iii) Indeed, the authors write "The absence of FANCD2 induces a mitochondrial stress response, similar to the ISRmt, involving the ATF4 pathway and a UBL5-dependent mtUPR, which cannot be correctly modulated and coordinated with replication, generating transcription-associated replication stress and genome instability." This statement nearly exactly entails what I would like to convey to the authors in that what they observe is a "mitochondrial stress response, similar to the ISRmt". This should be the main message of the paper/abstract/model. Despite of the lack of data on a UPRmt induction, the link to UBL5 and possibly UPRmt is reasonable to be made, but cannot be the focus of the paper due to lack of supporting data. If the authors were to change title/manuscript/model accordingly, I would strongly suggest publication.

<<Again, I agree with the reviewer due to the issues raised in ii). The association between UBL5 is nonetheless intriguing but cannot be used as a focus for the paper.

Please find below (in blue) a point by point response to the reviewers' comments.

Reviewers' comments:

Reviewer #3 (Remarks to the Author):

Experimentally, I think it is a good manuscript. It remains hard to follow why the authors do not change their wording (and some interpretation) to what is standard in the proteostasis field. For me as a proteostasis researcher, wrong semantics and concepts make this manuscript very hard to read and to follow the conclusions as stated:

i) In the abstract and general text, the authors still write UPR to refer to mitochondrial unfolded protein response. UPR is a very well established abbreviation that always means ER stress (never mitochondrial stress). It cannot be used for mitochondrial UPR without adding a nominator such as mt (i.e. mtUPR or UPRmt) as everyone in the field will otherwise be confused (as am I).

We apologize for this. We have now corrected or specified when we refer to mitochondrial UPR or ER stress.

ii) The authors write "We have then taken into account the point raised by the reviewer on the fact that chaperonin expression is an essential marker for mtUPR induction. In fact, we could not find induction of these genes at the mRNA level after depletion of FANCD2 in our cells." Thus, the authors do not see induction of the UPRmt! If the markers of the stress are not induced, by general definition in the field, the stress is not induced. It appears irrelevant in this context what effects are described in other papers for studies of mice with mutations in different genes that directly lead to mitochondrial misfolding. I would advise the authors to refer to ISR(mt) or mitochondrial stress response rather than UPRmt as UPRmt markers are not shown to change.

We meant here that the lack of induction of mtUPR markers in absence of FANCD2 can be due to a tissue-specific or time dependent nature of the response but also to the fact that FANCD2 itself may regulate chaperonin expression since it binds to their promoter. Notwithstanding, we have now referred to mitochondrial stress response the response observed in absence of FANCD2.

iii) Indeed, the authors write "The absence of FANCD2 induces a mitochondrial stress response, similar to the ISRmt, involving the ATF4 pathway and a UBL5-dependent mtUPR, which cannot be correctly modulated and coordinated with replication, generating transcription-associated replication stress and genome instability." This statement nearly exactly entails what I would like to convey to the authors in that what they observe is a "mitochondrial stress response, similar to the ISRmt". This should be the main message of the paper/abstract/model. Despite of the lack of data on a UPRmt induction, the link to UBL5 and possibly UPRmt is reasonable to be made, but cannot be the focus of the paper due to lack of supporting data. If the authors were to change title/manuscript/model accordingly, I would strongly suggest publication.

Following the reviewer suggestion, we have referred to a mitochondrial stress response, similar to the ISRmt, involving the ATF4 pathway and a UBL5-dependent mtUPR. We have done our best to change the title, abstract, text and model accordingly.

Reviewer #4

I've read over the manuscript by Fernandes et al. and I tend to agree with Reviewer #3:

i) In the abstract and general text, the authors still write UPR to refer to mitochondrial unfolded protein response. UPR is a very well established abbreviation that always means ER stress (never mitochondrial stress). It cannot be used for mitochondrial UPR without adding a nominator such as mt (i.e. mtUPR or UPRmt) as everyone in the field will otherwise be confused (as am I).

<< I too found the use of "UPR" confusing. The precise use of "UPR" historically refers to the stress response pathway originating from ER stress. The UPR resulting from mitochondrial stress must be referred to as mitochondrial UPR or an abbreviated form (e.g. UPRmt).

We apologize for this confusion, we have corrected it.

ii) The authors write "We have then taken into account the point raised by the reviewer on the fact that chaperonin expression is an essential marker for mtUPR induction. In fact, we could not find induction of these genes at the mRNA level after depletion of FANCD2 in our cells." Thus, the authors do not see induction of the UPRmt! If the markers of the stress are not induced, by general definition in the field, the stress is not induced. It appears irrelevant in this context what effects are described in other papers for studies of mice with mutations in different genes that directly lead to mitochondrial misfolding. I would advise the authors to refer to ISR(mt) or mitochondrial stress response rather than UPRmt as UPRmt markers are not shown to change.

<< Indeed, the UPRmt is classically associated with induction of these genes at the mRNA level. I agree with the reviewer to refer their findings as an ISRmt or mitochondrial stress response rather than the UPRmt.

See response to reviewer #3.

iii) Indeed, the authors write "The absence of FANCD2 induces a mitochondrial stress response, similar to the ISRmt, involving the ATF4 pathway and a UBL5-dependent mtUPR, which cannot be correctly modulated and coordinated with replication, generating transcription-associated replication stress and genome instability." This statement nearly exactly entails what I would like to convey to the authors in that what they observe is a "mitochondrial stress response, similar to the ISRmt". This should be the main message of the paper/abstract/model. Despite of the lack of data on a UPRmt induction, the link to UBL5 and possibly UPRmt is reasonable to be made, but cannot be the focus of the paper due to lack of supporting data. If the authors were to change title/manuscript/model accordingly, I would strongly suggest publication.

<<Again, I agree with the reviewer due to the issues raised in ii). The association between UBL5 is nonetheless intriguing but cannot be used as a focus for the paper.

See response to reviewer #3, we have modified the paper accordingly.